# PEEKING INSIDE THE BLACK-BOX: REINFORCEMENT LEARNING FOR EXPLAINABLE AND ACCURATE RELATION EXTRACTION

## ABSTRACT

This paper introduces a framework for relation extraction (RE) that enhances both accuracy and explainability. The framework has two key components: (i) a reasoning mechanism that formulates relation extraction as a series of text-processing steps inspired by cognitive science, and (ii) an optimization process driven by reinforcement learning (RL) with a novel reward function designed to improve both task accuracy and explanation quality. We call our approach COGRE. Our framework addresses the lack of supervision for language-based explanations in traditional RE by promoting outputs that include important relation keywords. These keywords are drawn from a high-quality dictionary that is automatically constructed using an LLM. We evaluate our approach for the task of one-shot RE using two LLMs and two RE datasets. Our experiments show that COGRE improves explanation quality by addressing two common failure patterns in one-shot RE: poor attention focus and limited one-shot learning capability. For example, our cognitive-structured reasoning with Qwen2.5-15B-Instruct on One-shot NYT29 achieves 24.65% F1, surpassing prior reasoning-based designs. Optimizing this approach with RL using our reward further improves performance by +23.46% (absolute). Finally, human evaluation shows that our best model generates relational keywords closely aligned with gold labels, increasing human explanation quality ratings by 54% (relative). Anonymous Github

## 1 INTRODUCTION

Relation extraction (RE), the natural language processing task that identifies relations between entities in text (Zelenko et al., 2003; Bunescu & Mooney, 2005), has been widely applied as a fundamental task in high-stakes domains where explainability is important such as healthcare, law, and finance (Adadi & Berrada, 2018; Goodman & Flaxman, 2017). However, previous RE methods that rely on feature-based models (Kambhatla, 2004), neural network architectures (Zeng et al., 2014), or more recently, pre-trained small language models (Soares et al., 2019; Sabo et al., 2021; Vacareanu et al., 2024a) still suffer from (1) limited explainability (Rosenman et al., 2020; Taillé et al., 2021), and (2) in some cases, the need for handcrafted training datasets that are expensive to annotate. All these issues impact the rapid and robust deployment of RE applications in critical domains.

Therefore, to build an RE system with improved generalization and explainability that can be rapidly customized and deployed, this work studies a variant of the one-shot RE task (Han et al., 2018) in which, given only a support sentence for each relation, models are required not only to extract relations but also to generate explanations for why such extractions are made.

Recently, large language models (LLMs) have demonstrated strong language understanding and reasoning abilities (Ahn et al., 2024; Luo et al., 2024), which inspires us to adopt LLMs for the RE task. However, it is known that "LLMs do not say what they think" (Turpin et al., 2023; Liu et al., 2025), i.e., their explanations do not faithfully align with their decisions. To mitigate this limitation, we propose a cognitive-structured framework for relation extraction (COGRE) that jointly optimizes task accuracy and explainability. Our approach mimics how humans process complex textual input: cognition emerges not from storing sequential words in limited memory slots (Miller, 1956), but from a construction–integration process that yields a coherent logical chain (Kintsch,

1988). More formally, our framework formulates RE into three steps: (i) chunking from text into logical propositions; (ii) anchoring certain keywords as cues; and (iii) integrating these cues through a verbalized explanation. We optimize this framework with reinforcement learning (RL) with a novel reward mechanism that *jointly* judges task accuracy and the quality of the corresponding explanation. Because we do not have supervision on the latter component, we approximate it using a method that matches explanation cues with a credit dictionary constructed from high-quality, self-generated explanations produced by an LLM. We call our explanation-level reward HIT@DICT.

Our specific contributions are driven by the following questions:

(1) *What is a reliable framework for LLMs to perform RE reasoning?* We propose a RE method that is loosely inspired by structured cognition (see Related Work 2.3). Our framework decomposes the RE task into three steps: (i) semantic chunking; (ii) keyword anchoring; (iii) integrative reasoning. This bottom-up design reduces LLMs' processing burden and mitigates reasoning hallucinations during analyzes of complex sentences.

(2) *How to design a reward that jointly supervises accuracy and reasoning quality in RE task?* We design HIT@DICT reward, a simple rule-based reward mechanism. We sample true positive outputs from a "vanilla" LLM. Given these outputs paired with their respective relation label, we use GPT-4o to extract relational keywords from each data point to construct a credit dictionary. During RL training, the credit dictionary is used to assign rewards by counting the occurrences of these dictionary items in the model's outputs. Thus, the HIT@DICT reward offers a fine-grained signal that reinforces the model's own reasoning behaviors without relying on human-filtered references.

(3) *How to evaluate a RE system that balances accuracy and explainability?* We introduce a dual evaluation method that combines both automatic evaluation and human evaluation on explanation quality, filling the often-overlooked gap of explanation in RE. Our proposed COGRE surpasses strong RE baselines, e.g., achieving F1 score of 31.06% and 24.65% in one-shot TACRED and NYT29 (Alam et al., 2024), respectively. With HIT@DICT, reinforcement learning further improves F1 score by 37.31% and 48.11%. Importantly, our method improves human rating of explanation quality by 24.72% and 54.24% (relative).

(4) *What are the primary failure modes of LLMs in relation extraction?* Our error analysis identifies the main failure of "vanilla" LLMs on the RE task is mismatching the abstraction level of inferred relations with RE annotations granularity; for example, LLMs struggle to distinguish geographic scale in `org:city_of_headquarters`. We conduct a human analysis of explanations generated by the Phi-4 before and after trained with both accuracy and HIT@DICT reward. The trained version produces more concise summaries in 20% cases and shows better alignment with RE labeling in 37.5% cases. In more detail, the trained model tends to include relational keywords closely aligned with gold labels in their explanations (e.g., *enroll*, *attend*, and *university* for the relation `per:schools_attended`), while the untrained model uses vague terms such as *associated* or *institution*.

## 2 RELATED WORK

### 2.1 EXPLAINABLE RELATION EXTRACTION.

Relation extraction is widely applied in high-stakes domains such as healthcare, law, and finance (Adadi & Berrada, 2018; Goodman & Flaxman, 2017), where explainability is critical. Traditional RE models, including feature-based methods, neural networks, and pre-trained small language models, attempt to provide explainability through attention weights (Zhou et al., 2016), feature importance (Kambhatla, 2004), or post-hoc analysis (Wickramasinghe et al., 2021). In parallel, rule-based methods enable transparent model adjustment (Vacareanu et al., 2024b; Tang & Surdeanu, 2023) and inspire our symbolic reward for RL training. However, due to the lack of language-based explanations, these approaches have limited explainability.

### 2.2 LLM REASONING.

Explicit reasoning in LLMs enhances explainability via human-readable traces (Wei et al., 2022; Chu et al., 2025) and improves downstream accuracy through prompting strategies such as least-

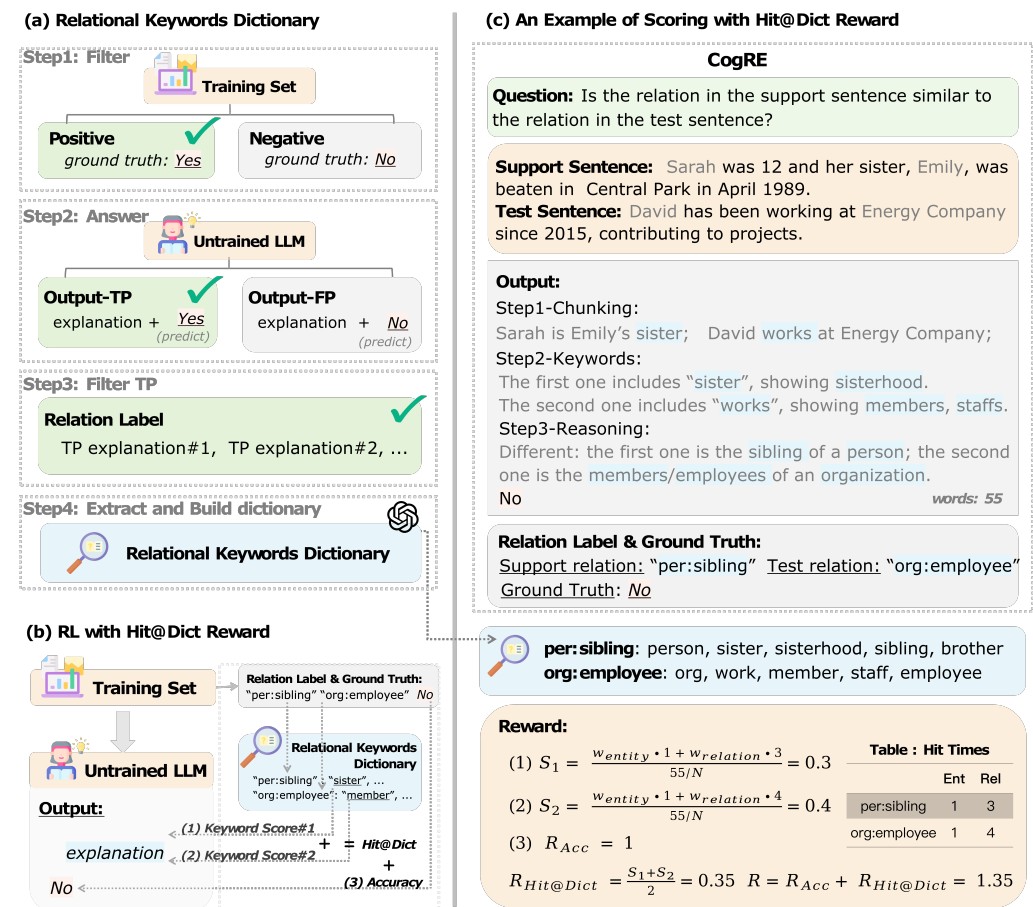

Figure 1: An overview of the COGRE framework. (a) Relational Keywords Dictionary: relational keywords are extracted from explanations of true positive samples generated by untrained LLMs to build a dictionary (Alg. 1). (b) Reinforcement Learning with HIT@DICT: LLM outputs scored by accuracy (answers) and HIT@DICT (explanations). (c) Example of Scoring with HIT@DICT: COGRE enables stepwise reasoning. Keywords in the dictionary are matched against the LLM output (Hit Times Table); the HIT@DICT reward counts a normalized hit rate (Section 3.2).

to-most (Zhou et al., 2023). For RE, recent work leverages LLMs via few-shot prompting (Wan et al., 2023; Ma et al., 2023) and instruction tuning (Ouyang et al., 2022; Qi et al., 2024). However, LLMs often generate explanations of limited quality (Turpin et al., 2023; Liu et al., 2025). Recently, reinforcement learning with verifiable rewards improves accuracy and explainability (He et al., 2025). However, explanation-oriented rewards remain limitedly explored. Existing methods rely either on simple format signals (Wen et al., 2025; Xin et al., 2025) or costly LLM-as-a-judge approaches (Saha et al., 2025; Huang et al., 2025). In this work, we propose a fine-grained RL reward that enhances both accuracy and explainability in RE.

## 2.3 HINTS FROM COGNITIVE PSYCHOLOGY.

Existing work shows that cognitive psychology provides useful insights for LLMs and evidences their cognitive capabilities (Yax et al., 2024; Niu et al., 2024). Cognitive psychology has also extensively studied how humans process information. The Construct-Integration model describes comprehension in four steps: forming concepts, elaborating, inferring new propositions, and integrating them into a representation (Kintsch, 1988). Several separate studies also show: *chunking* reduces cognitive load (Miller, 1956), *keyword anchors* guide attention (Kintsch & Van Dijk, 1978), and cognitive monitoring improves strategy adjustment (Flavell, 1979). Motivated by these, we frame RE as a three-step framework mimicking human processing.

---

**Algorithm 1** Building the Relational Keywords Dictionary

---

**Require:** Training set $\mathcal{D}_{\text{train}} = \{(s_1, s_2, r_1, r_2, y)\}$; vanilla LLM $\mathcal{M}$; GPT-4o (or equivalent) API; sample size per label $K \in \{1, \ldots, 5\}$

**Ensure:** Keywords dictionary Dict : relation_label $\mapsto$ keywords list

1: $\mathcal{C} \leftarrow \{(s_1, s_2, r_1, r_2, y) \in \mathcal{D}_{\text{train}} \mid r_1 = r_2 \ \wedge \ y = \text{"Yes"}\}$    ▷ Pairs with identical relation labels
2: $\mathcal{G} \leftarrow \emptyset$    ▷ Good cases predicted correctly by the vanilla LLM
3: **for all** $(s_1, s_2, r_1, r_2, y) \in \mathcal{C}$ **do**
4:    $(\hat{z}, \hat{y}) \leftarrow \mathcal{M}(s_1, s_2)$    ▷ Vanilla LLM inference: explanation & label
5:    **if** $\hat{y} = \text{"Yes"}$ **then**
6:      $\mathcal{G} \leftarrow \mathcal{G} \cup \{(s_1, s_2, r_1, \hat{z})\}$    ▷ Keep good cases
7:    **end if**
8: **end for**
9: Group $\mathcal{G}$ by relation label: for each $r \in \mathcal{R}$, let $\mathcal{G}_r = \{(s_1, s_2, r, \hat{z}) \in \mathcal{G}\}$
10: Dict $\leftarrow \emptyset$
11: **for all** $r \in \mathcal{R}$ **do**
12:    $\mathcal{S}_r \leftarrow \text{SampleUpTo}(\mathcal{G}_r, K)$    ▷ Sample $1 \sim 5$ good cases per label
13:    $\text{prompt}_r \leftarrow \text{BuildPrompt}(r, \mathcal{S}_r)$    ▷ Prompt includes the label and several examples
14:    $\text{keywords}_r \leftarrow \text{GPT-4o}(\text{prompt}_r)$    ▷ Generate relational keywords list
15:    $\text{labelKeywords}_r \leftarrow \text{Tokenize}(\text{label}_r)$    ▷ Decompose the relation label into keywords
16:    $\text{Dict}[r] \leftarrow \text{PostProcess}(\text{keywords}_r \cup \text{labelKeywords}_r)$    ▷ Lowercasing, dedup, stemming/lemmatization, stopword removal
17: **end for**
18: **return** Dict

---

## 3 METHOD

Our pilot error analysis reveals (Section 5.3) that LLMs always conduct token-level matching between two sentences and overlook the semantics that truly convey the relation. To address this gap, we design a framework loosely inspired by cognitive science (Kintsch, 1988) to guide LLMs in analyzing core relations verbalized in natural language sentences.

### 3.1 COGNITIVE-STRUCTURED RE

As shown in Figure 1(c), our Cognitive-Structured RE (COGRE) framework formulates RE reasoning into three steps. First, *Proposition Chunking*, where the LLM summarizes each sentence into a relational proposition. This step ensures that the LLMs' analysis process starts with compressed propositions instead of long sequences of tokens. Next, *Keywords Anchoring*, where the LLMs anchor relational keywords in the input sentences and propositions, which grounds the LLMs' relation-matching reasoning in the original sentence and the extracted propositions. The final step is *Integrative Reasoning*. The LLMs are prompted to integrate propositions and keywords into a coherent logical chain. Formally, suppose the LLM $\mathcal{M}$ is parameterized by $\theta$. Let the input be $x = (s_1, s_2)$, where $s_1$ and $s_2$ are two input sentences. Given the input $x = (s_1, s_2)$, the LLM produces a readable explanation $\hat{z}$ followed by the final label $\hat{y}$, which can be formulated as:

$$(\hat{z}, \hat{y}) \sim \mathcal{M}_\theta(\cdot \mid s_1, s_2). \tag{1}$$

### 3.2 REINFORCEMENT LEARNING WITH HIT@DICT REWARD

Improving the quality of explanations without introducing an agentic reward is challenging in RL training, while format-based signals that ignore reasoning content provide only weak guidance for reasoning. Additionally, human-annotated rewards tend to induce annotator preferences that may deviate from the model's actual reasoning behavior (Xue et al., 2024). To overcome these limitations, we propose an efficient explanation reward, namely the HIT@DICT reward, and integrate it with the accuracy reward to incentivize LLMs for reliable RE reasoning. Intuitively, our reward promotes both task accuracy and high-quality explanations.

**Reward Function Design.** To provide effective training signals, we design the reward function with two complementary components. The first part is the HIT@DICT reward, which evaluates the

occurrences of these relational keywords in the LLMs explanation based on the predefined credit dictionary. The second part is the accuracy reward, which directly evaluates the correctness of the predicted results. Together, these two components define the final reward:

$$\mathcal{R} = \mathcal{R}_{\text{Acc}} + \mathcal{R}_{\text{HIT@DICT}}. \tag{2}$$

Here, $\mathcal{R}_{\text{Acc}}$ is the accuracy reward, while $\mathcal{R}_{\text{HIT@DICT}}$ is the rule-based explanation reward. This formulation ensures that the model is incentivized for both correct predictions and explanations that align well with relational knowledge.

**HIT@DICT Reward.** As shown in Figure 1, the relational keywords dictionary serves as a core component of our framework. It collects all relation labels appearing in the training dataset, together with their associated relational keywords. Unlike human-crafted keyword lists, these keywords are automatically derived from the outputs of vanilla LLMs. Importantly, this process happens offline, so the inference overhead is minimal.

*How to construct a Relational Keywords Dictionary?* We sample all the positive items where the support sentence and the test sentence share the same relation label. Then, the vanilla model answers all these positive items, and we filter the true positive items with the final answer, "Yes". For each label that appears in the training dataset, we sample one to five LLM-generated explanations. These relation labels, combined with their associated explanation cases, are input into GPT-4o. GPT-4o extract the relational keywords from these cases. Additionally, each relation label is decomposed into keyword tokens as part of the relational keywords. After text post-processing, these relation labels and their associated keywords are added to the relation keywords dictionary. The detailed algorithm is illustrated in Alg. 1; we show a simple example in Figure 1.

*How can the* HIT@DICT *reward be applied?* For input sentences $(s_1, s_2)$, the HIT@DICT reward measures how many relational keywords in $\hat{z}$ match the relational keywords dictionary. Given an explanation $\hat{z}$ and a relation label $r$, we compute the HIT@DICT score as follows. Let $\text{Entity}(r)$ denotes the set of entity-related keywords and $\text{Rel}(r)$ the set of relational keywords associated with $r$. We define the weighted hit counts as:

$$\mathcal{H}_{\text{entity}}(\hat{z}, r) = \sum_{k \in \text{Entity}(r)} \mathbf{1}[k \in \hat{z}], \qquad \mathcal{H}_{\text{relation}}(\hat{z}, r) = \sum_{k \in \text{Rel}(r)} \mathbf{1}[k \in \hat{z}], \tag{3}$$

where $\mathbf{1}[\cdot]$ is an indicator function that equals 1 if keyword $k$ appears in $\hat{z}$, and 0 otherwise. For the special case $r = \texttt{no\_relation}$, we set $\mathcal{H}_{\text{entity}} = \mathcal{H}_{\text{relation}}$. The total weighted hits are given by:

$$\mathcal{H}(\hat{z}, r) = w_{\text{entity}} \cdot \mathcal{H}_{\text{entity}}(\hat{z}, r) + w_{\text{relation}} \cdot \mathcal{H}_{\text{relation}}(\hat{z}, r), \tag{4}$$

with two hyper parameters $w_{\text{entity}}$ and $w_{\text{relation}}$. Let $|\hat{z}|$ denote the number of words in $\hat{z}$, normalized by a factor of $N$ (a third hyper parameter). The final score is defined as:

$$\mathcal{S}(\hat{z}, r) = \frac{\mathcal{H}(\hat{z}, r)}{|\hat{z}|/N} \tag{5}$$

Finally, the overall HIT@DICT *reward* aggregates the contributions from both sentences $s_1$ and $s_2$, calculated as $\mathcal{R}_{\text{HIT@DICT}} = (\mathcal{S}(\hat{z}, r_1) + \mathcal{S}(\hat{z}, r_2))/2$. Here $r_1$ and $r_2$ denote the ground truth relation labels for $s_1$ and $s_2$, respectively. An example of scoring with HIT@DICT reward is provided in Figure 1(c). In this case, the hit times (seen in Table: Hit Times) of entities and relations are used to compute partial scores $S_1$ and $S_2$. The final $R_{\text{HIT@DICT}} = (S_1 + S_2)/2 = 0.35$.

**Accuracy Reward.** We introduce the accuracy reward $\mathcal{R}_{\text{Acc}}(\hat{y}, y)$ to evaluate the correctness of the predicted label. In one-shot settings, each test sentence is matched with $K$ supports, with at most one positive and the rest negative, leading to a 1:$K$ imbalance. Following (Lin et al., 2019), to counter this, we weigh the reward by assigning higher scores to correct Yes predictions and stronger penalties to incorrect Yes predictions, encouraging the model to align with the task's inherent imbalance:

$$\mathcal{R}_{\text{Acc}}(\hat{y}, y) = \begin{cases} 3.0, & \text{if } \hat{y} = \text{Yes} \wedge y = \text{Yes} \\ 1.0, & \text{if } \hat{y} = \text{No} \wedge y = \text{No} \\ -3.0, & \text{if } \hat{y} = \text{Yes} \wedge y = \text{No} \\ -1.0, & \text{if } \hat{y} = \text{No} \wedge y = \text{Yes} \\ 0.0, & \text{otherwise} \end{cases} \tag{6}$$

**Training Process.** We optimize COGRE with Group Relative Policy Optimization (GRPO) (Shao et al., 2024). Formally, given a group of $m$ explanation and label pairs $\mathcal{O} = \{(\hat{z}_i, \hat{y}_i) \mid i \in [m]\}$ sampled from COGRE for the same input $(s_1, s_2)$, we assign each pair a scalar reward $R(\hat{z}_i, \hat{y}_i)$ using our designed function $\mathcal{R} = \mathcal{R}_{\text{Acc}} + \mathcal{R}_{\text{HIT@DICT}}$. GRPO encourages relative improvements within a group by normalizing each reward against the group mean. Specifically, the *group-relative advantage* of the $i$-th explanation–label pair is defined as:

$$\mathcal{A}_i = \frac{R(\hat{z}_i, \hat{y}_i) - \frac{1}{m} \sum_{j=1}^{m} R(\hat{z}_j, \hat{y}_j)}{\text{std}(\{R(\hat{z}_j, \hat{y}_j) \mid j \in [m]\})}, \tag{7}$$

where $\text{std}(\cdot)$ is the standard deviation of group rewards. The overall GRPO objective is optimized to maximize a clipped function with a KL penalty:

$$\mathcal{L}(\theta) = \mathbb{E}_{(\hat{z}_i, \hat{y}_i) \sim \mathcal{O}} \left[ \min(\rho_i \, \mathcal{A}_i, \, \text{clip}(\rho_i, 1 - \epsilon, 1 + \epsilon) \, \mathcal{A}_i) - \beta \, \text{KL}(\theta \,\|\, \theta_{\text{ref}}) \right], \tag{8}$$

where $\rho_i$ is the importance ratio between the updated and old policy probabilities, $\epsilon$ controls the clipping range, and $\beta$ weights the penalty for diverging from a reference model $\theta_{\text{ref}}$.

## 4 EXPERIMENT SETUP

**Benchmark.** We conduct experiments on two datasets, i.e., *Few-shot TACRED* and *NYT29* (Alam et al., 2024), in one-shot setting. Notably, the relation labels in the training partition and the testing partition are out-of-distribution. Besides, since traditional RE methods typically rely on small classifiers, RE benchmarks are built to be extremely large. Following previous work (Li et al., 2023), we also randomly sampled 1,000 episodes for each partition according to the original proportions of each relation label. We provide the statistics of the sampled test datasets in Appendix A.6.

**Evaluation.** We adopt a dual evaluation protocol on both automatic and human evaluation. We use the F1 score as the automatic evaluation metric, which computes task accuracy. For human evaluation, we rated explanations on a 3-point Likert scale: two points for the correctness and conciseness of the two summaries, plus one point if the abstraction level aligns with RE labeling. The detailed evaluation rubric is provided in Appendix A.2. Two annotators with NLP backgrounds rated the sampled explanations independently. The Cohen's kappa score is 0.693, indicating substantial agreement and that our evaluation rubric is well-defined.

**Baselines.** We compare our method with two categories of baselines: RE prompting strategies and conventional supervised RE models. *Prompting RE baselines*: (i) SUMASK (Li et al., 2023) reformulates relation extraction as a multi-turn question answering task. We implement the original and a one-prompt variant of SUMASK (multi-turn interactions merged into a single prompt; see Appendix A.10), reporting only the latter due to its consistently stronger performance. (ii) Naive prompting: two simple variants—*direct-matching* (outputs "Yes"/"No") and *simple-reasoning* (produces reasoning before "Yes"/"No"). See *Conventional RE Models*: Semantic Rule Matcher (Vacareanu et al., 2024b), which combines a neural classifier with rules, achieving state-of-the-art results on Few-Shot TACRED and NYT29.

**Implementation Details.** We sample 20,000 items from the training partition,preserving the distribution of relation labels and maintaining an approximate 1:7 ratio between positive and negative instances (statistics in Appendix A.5). We implement our method with Qwen-2.5-14B-instruct and Phi-4, using fixed reward hyperparameters: $N = 5$, $w_{\text{entity}} = 0.4$, and $w_{\text{relation}} = 1.0$. These values were heuristically chosen and kept constant across all experiments. We optimize the model using Verl (Sheng et al., 2025) with an actor learning rate of $1 \times 10^{-6}$, KL regularization (coefficient 0.01), and entropy regularization (coefficient 0.001). Training is conducted on 4×NVIDIA-H100-80GB GPUs. A complete run on the 14B–15B model takes 20 GPU-hours.

## 5 EXPERIMENT RESULTS

### 5.1 MAIN RESULTS

We present our main results in Table 1. We focus on comparing different reasoning designs and the impact of RL with only accuracy rewards, and with both HIT@DICT and accuracy rewards. We draw the following two main observations from these experiments:

| Method | One-shot TACRED | | | One-shot NYT29 | | |
|---|---|---|---|---|---|---|
| | Prec% | Recall% | F1% | Prec% | Recall% | F1% |
| ***Baselines*** | | | | | | |
| - Semantic Rule Matcher | 32.45 | 19.72 | 24.52 | 22.23 | 13.45 | 16.76 |
| - SUMASK *(Phi-4)* | 4.44 | 31.71 | 7.78 | 10.96 | 26.13 | 15.44 |
| ***Phi-4*** | | | | | | |
| ***Before RL*** | | | | | | |
| - Direct Matching | 5.43 | 26.83 | 9.03 | 9.04 | 23.15 | 13.00 |
| - Simple Reasoning | 5.69 | 58.54 | 10.38 | 8.71 | 30.68 | 13.57 |
| - COGRE *(our)* | 22.53 | 50.00 | 31.06 | 12.03 | 30.68 | 17.28 |
| ***After RL with Acc*** | | | | | | |
| - COGRE *(our)* | 26.90 | 47.56 | 34.36 | 20.45 | 40 | 41.02 |
| ***After RL with HIT@DICT + Acc*** | | | | | | |
| - COGRE *(our)* | 26.88 | 60.98 | 37.31 | 45.14 | 44.89 | 45.01 |
| ***Qwen2.5-14B-Instruct*** | | | | | | |
| ***Before RL*** | | | | | | |
| - Direct Matching | 13.33 | 2.44 | 4.12 | 48.73 | 13.63 | 21.31 |
| - Simple Reasoning | 5.67 | 34.15 | 9.72 | 11.85 | 34.23 | 17.61 |
| - COGRE *(our)* | 29.49 | 28.05 | 28.75 | 20.18 | 31.67 | 24.65 |
| ***After RL with Acc*** | | | | | | |
| - COGRE *(our)* | 26.83 | 40.24 | 32.20 | 26.17 | 29.40 | 27.69 |
| ***After RL with HIT@DICT + Acc*** | | | | | | |
| - COGRE *(our)* | 22.08 | 62.20 | 32.58 | 63.34 | 38.78 | 48.11 |

Table 1: Precision (P), recall (R), and F1 on the one-shot TACRED and one-shot NYT29 datasets. We split the table into three blocks: baseline methods, vanilla prompting methods before reinforcement learning, and after reinforcement learning with accuracy reward, and with both HIT@DICT and accuracy reward. Green highlights F1 scores, with darker shades indicating larger values.

**COGRE improves accuracy with balanced precision and recall.** Our COGRE consistently outperforms all baselines with higher F1 and more balanced precision and recall. In contrast, the Semantic Rule Matcher (the previous SOTA), based on rules and a small language model with poorer generalization, yields relatively high precision but lower recall. Prompting-based LLMs baselines rely solely on LLMs' generalization ability, leading to strong recall but lower precision. Our COGRE combines both perspectives: it anchors reasoning with rule-based keywords while leveraging LLMs' generalization through summarization and integrative reasoning.

**Outcome reward improves task accuracy.** As shown in Table 1, Qwen2.5-14B-Instruct, trained only with the accuracy reward, surpasses its non-trained backbone by +3.45% and +23.74%, while the trained Phi-4 leads to +3.30% and +3.04% improvements, respectively. The HIT@DICT + Acc reward further boosts accuracy across models, outperforming accuracy-only training and reaching state-of-the-art. In particular, Qwen2.5-14B-Instruct reaches 48.11% F1 with HIT@DICT + Acc, a 73.74% relative gain over the accuracy-only method. Moreover, we further investigate the quality of the explanations generated by LLMs trained with the HIT@DICT + Acc reward (see Section 5.3).

## 5.2 BEHAVIOR OF HIT@DICT REWARD

We further monitor the RL training process of the models and compare the impact of the accuracy reward and HIT@DICT reward. Figure 2 shows the RL training process of both Phi-4 and Qwen2.5-14B-Instruct on the one-shot NYT29 dataset.

**HIT@DICT reward accelerates the convergence of training.** It can be seen in Figure 2 that the convergence points of the reward curves (e.g., a and d) and the reward–KL penalty curves (e.g., d and e) differ. In Figure 2(a), the HIT@DICT +*Acc* curve rapidly increases to above 1.1 within 5 hours and stabilizes between 1.1–1.2 after 10 hours; in contrast, the *Only Acc* curve climbs slowly

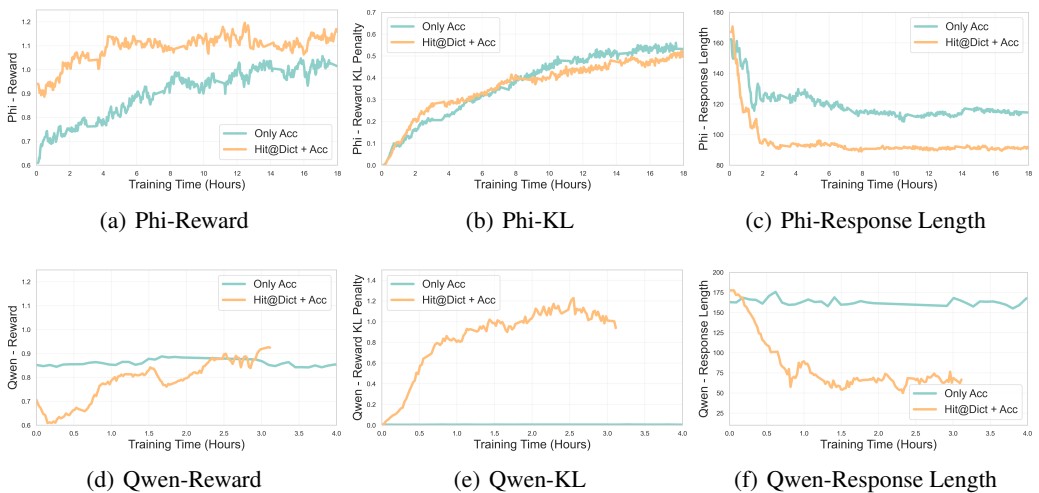

(a) Phi-Reward          (b) Phi-KL          (c) Phi-Response Length

(d) Qwen-Reward          (e) Qwen-KL          (f) Qwen-Response Length

Figure 2: Training dynamics on the one-shot NYT29 dataset for Phi-4 and Qwen2.5-14B-Instruct. The Y-axes show reward, KL penalty, and response length. We compare reinforcement learning with accuracy reward *Only Acc* and with the combined HIT@DICT reward HIT@DICT *+Acc* .

from 0.6 and stabilizes around 1.0 after 13 hours. In Figure 2(b), the HIT@DICT *+Acc* curve also quickly rises from 0.7 to above 0.95 within 2 hours and then levels off, whereas the *Only Acc* curve remains around 0.5–0.6 with no clear sign of convergence. Similar patterns are observed in the reward–KL penalty curves in Figure 2(b) and (e). All in all, this analysis shows that training with the HIT@DICT reward consistently converges faster than training with the accuracy reward alone.

**HIT@DICT reward extends the capability boundary of RL in improving models.** As shown by the final reward values in Figure 2 (a) and (d), and the accuracy results in Table 1, training with the HIT@DICT reward yields higher values. For example, the HIT@DICT *+Acc* curves achieve higher final reward values than the *Only Acc* curves, by approximately +0.15 and +0.06. This shows that training with HIT@DICT rewards allows RL to further extend the boundaries of model capability.

**HIT@DICT reward provides more stable training.** As shown in Figure 2 (d) and (e), Qwen2.5-14B-Instruct, trained only with the accuracy reward, exhibits stagnant reward values and an extremely small reward–KL penalty, indicating the policy remains almost unchanged from its initial state. In contrast, Qwen2.5-14B-Instruct, trained with the HIT@DICT reward, shows steady growth. This indicates that the HIT@DICT reward provides a more stable and effective learning signal.

**HIT@DICT reward encourages more concise explanations.** As Figure 2 (c) and (f) show, with the HIT@DICT reward, the response length compresses to 75–90 tokens. Combined with our analysis of the explanations, we found that in most cases, the models produce more concise and accurate stepwise explanations. It enhances reasoning efficiency with more concise outputs. However, we also observe a trade-off between response length and explainability: in the setting of training Qwen2.5-14B-instruct on the NYT29, the model skips the reasoning after chunking in some cases.

## 5.3 ERROR ANALYSIS OF LLM ON RE-REASONING

**Simple reasoning strategy.** We analyze the explanations of vanilla LLMs using a random reasoning strategy. For this stage, we select Qwen2.5-14B-Instruct and GPT-4o. From their explanations, we identify two common failure patterns. *First*, failing to focus on semantics that truly convey relation. When matching two sentences, LLMs frequently focus on irrelevant tokens in the second sentence, aiming to align

| Gold Label | *per:alternate_name*    *no_relation* |
|---|---|
| **Support Sentence** | **Gadahn** is also known as **Azzam al-Amriki**. |
| **Test Sentence** | **Arcandor** was known as Karstadt in **2000**. |

Table 2: An example case for the failure pattern. The bold entities highlight the two entities between which the relation should be extracted.

with the relation conveyed in the first. We pro-
vide a simplified example in Table 2. In this case, LLMs incorrectly focus on two names in the second sentence in order to mimic the relation of `per:alternate_name` in the first sentence. *Second*, failing to align with the abstraction level defined in the RE human-annotation schema. Without the human-crafted descriptions of relation labels, LLMs struggle to distinguish between similar human-defined relations, e.g., `org:country_of_headquarters` and `org:city_of_headquarters`. It's also a common challenge in one-shot and few-shot RE.

**COGRE.** Then, we evaluate the quality of LLM explanations at three stages: (i) vanilla LLMs with the COGRE framework, and after RL training, (ii) with only accuracy reward and (iii) with both HIT@DICT and accuracy reward. We select Qwen2.5-14B-Instruct and Phi-4 across two datasets. For each LLM–dataset–stage combination, we sample 40 explanations, with 10 explanations per category (TP, TN, FP, FN). Results (Appendix A.3) show that human evaluation scores improve by 54.24% (relative). Compared with vanilla and accuracy-only models, HIT@DICT combined with accuracy reward enables more concise summaries and better alignment with human annotations. For example, in the Phi–TACRED setting, among the 40 analyzed cases, the model trained with the HIT@DICT reward produced more concise summaries in 8 cases and exhibited better alignment with human labeling in 15 cases. In more detail, the trained model tends to include relational keywords closely aligned with gold labels in their explanation (e.g., *enroll*, *attend*, and *university* for the relation `per:schools_attended`), while the untrained model often relies on vague terms such as *associated* or *institution*. We provide some case comparisons in Appendix A.4.

## 5.4 ABLATION EXPERIMENTS

We analyze the effectiveness of each step in our COGRE framework. In each variant, one step is removed while the others remain:

(i) *w/o chunking*: Removes the step of chunking; (ii) *w/o keywords*: Removes the step of keywords anchoring. (iii) *w/o reasoning*: Removes the reasoning component. Experiment results on the one-shot TACRED and NYT29 datasets are reported in Table 3.

We highlight three key observations. *First*, all three steps contribute to the final performance. Removing any step from our framework leads to a clear performance drop, ranging

| Technique | One-shot TACRED | | | One-shot NYT29 | | |
|---|---|---|---|---|---|---|
| | P% | R% | F1% | P% | R% | F1% |
| COGRE | 22.53 | 50.00 | 31.06 | 12.03 | 30.68 | 17.28 |
| - *w/o chunking* | 7.20 | 5.73 | 12.79 | 8.20 | 27.41 | 12.63 |
| - *w/o keywords* | 16.10 | 52.44 | 24.64 | 10.39 | 31.39 | 15.62 |
| - *w/o reasoning* | 5.94 | 28.05 | 9.81 | 9.91 | 24.57 | 14.12 |

Table 3: Ablation study on Phi-4 across one-shot TACRED and NYT29, reporting precision (P), recall (R), and F1. Green highlights F1 scores, with darker shades indicating larger values.

from –1.66% to –21.51%. *Second*, the keywords anchoring step primarily contributes to precision. It is the only setting where recall increases (+2.44% and +0.71%) while precision decreases (–6.43% and –1.64%). *Third*, the chunking and reasoning steps support both precision and recall, but their impact is more apparent on recall. When these steps are removed individually, the recall decreases more substantially (-44.27% to -21.95% on TACRED; –3.27% to 6.11% on NYT29).

## 6 CONCLUSION

In this work, we introduced COGRE, a relation extraction framework loosely inspired by structured cognition. By decomposing RE into three steps—semantic chunking, keyword anchoring, and integrative reasoning—our approach reduces the processing burden of LLMs and mitigates reasoning hallucinations in complex sentences. To further enhance reasoning and explanation quality, we proposed the HIT@DICT reward, a lightweight reward that enables joint evaluation of task accuracy and explanation quality through a credit dictionary derived from self-generated explanations. Extensive experiments and human evaluations on one-shot TACRED and NYT29 demonstrate that our framework achieves enhanced accuracy and explanation quality. Human analysis confirms that our reward design encourages models to generate more concise and label-aligned reasoning.

## ETHICS STATEMENT

The research focuses on the development of a reasoning-augmented relation extraction framework. The proposed method enables Large Language Models (LLMs) to process complex sentences and compare relations through three-step reasoning. Then, we design a novel reward function to provide reward signals for reasoning quality. This work does not involve human subjects research beyond standard human evaluations. Our human-like reasoning design is a computational framework inspired by cognitive theories and does not involve human experiments. The human evaluation was conducted by two annotators with NLP backgrounds who voluntarily participated and did not provide any personal information. All datasets used in this study (one-shot TACRED and NYT29) are publicly available and widely adopted in prior work. No part of this work includes deceptive practices or intentional misuse of information. We are committed to conducting and presenting this research with integrity and social responsibility. We do not foresee any direct ethical risks or misuse beyond those already present in large language models.

## REPRODUCIBILITY STATEMENT

We provide the sampled training data, the full source code, and instructions to reproduce all experiments at `https://anonymous.4open.science/r/CogRE-Hit-at-Dict-4E75`. We also release the source code and data in the supplementary material.

Details of hyperparameters and hardware training setup are provided in Section 4.

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

# A  APPENDIX

## A.1  LLM USAGE STATEMENT

In this work, large language models (LLMs) were only used to polish the writing, such as improving grammar, readability. They were not involved in the conception of the research problem, the design of the methodology, the execution of experiments, or the analysis and interpretation of results. All substantive research contributions, including theoretical development, experimental design, and analysis, are solely the work of the authors, who take full responsibility for the content of this paper.

## A.2  HUMAN EVALUATION RUBRIC

**Main Rubric.**  To assess the quality of model-generated explanations, we adopt a human evaluation rubric that emphasizes both concise summarization and alignment with RE labeling. The rubric assigns scores based on three major criteria: (1) the correctness and conciseness of the summarization for the support sentence, ensuring that the key relational information is accurately captured without including irrelevant details; (2) the correctness and conciseness of the summarization for the test sentence, evaluated under the same principles; and (3) the alignment of the explanation with the labeling of whether the relations expressed in the two sentences match or not, with any abstraction error or illogical reasoning resulting in a deduction. Each explanation is scored on a 3-point scale, with details provided as follows.

---

*Human evaluation rubric for explanation quality (max score: 3).*

```
[1 point]
    A correct and concise summarization of the support sentence is
    awarded 1 point.

[1 point]
    A correct and concise summarization of the test sentence is awarded 1
     point.

[1 point]
    A reasonable explanation of whether relation_1 and relation_2 match
    or do not match is awarded 1 point. Any abstraction error results in
    the loss of this point.

 Special handling:
    - If summarization is incorrect but the explanation is logically
    reasonable, the third point can still be awarded.
    - Points are deducted when the model confuses similar relations,
      e.g., "city" vs. "country", or "reference" vs. "alternate_name".
```

---

**Common types of abstraction errors.**  In addition to the main rubric, we present several common types of abstraction errors to help graders develop a clearer understanding of how such errors should be identified and penalized. These error types serve as practical references to ensure consistent and fair scoring. The detailed description and illustrative examples are provided for each case as follows.

---

*Common Abstraction Errors for Human Rating*

```
Abstraction Error:
    If two sentences express the same relation, they must be abstracted
    in the same direction and at the same level.
    Example:
    ``He, 12-years-old, got a good offer.'' and ``Jam is 12.''
    - Correct: per:age; per:age  [0 points]
```

```
    - Incorrect: per:age; per:number  [-1 point]

several common types:
    - Lack of Higher-Level Deductive Abstraction:
        Description: When a higher-level deductive abstraction is
        required to align the relations, failing to apply it leads to
        error.
        Example:
            ``STX Finland is part of the international STX Europe Group''
            and
            ``Merck will acquire all of Millipore.''
            - Correct: org:parents; org:parents  [0 points]
            - Incorrect: org:parents; org:transaction  [-1 point]

    - Over- or Under-Focusing on Details:
        Description: The abstraction direction and level are correct, but
        the model misjudges due to being overly detailed or overly
        general.
        Example:
            ``The arrangement of financing for Millipore Corp in the US''
            and
            ``Burlington Northern Santa Fe Corp is the biggest bet yet on
            a US economic recovery.''
            - Correct: org:country_of_headquarters; org:
            country_of_headquarters  [0 points]
            - Incorrect: First = org:financial_transaction, Second = org:
            economic_event  [-1 point]
```

**Rubric Reliability Verification via Cohen's Kappa.** Furthermore, to verify the reliability of our rubric, we engaged two independent annotators. Both annotators had NLP research backgrounds. They first evaluated the explanations in the setting of Phi-4 on the one-shot TACRED dataset. Each annotator followed the rubric, scoring explanations based on the three criteria and abstraction error types. This independent evaluation enabled us to measure the consistency between annotators. Specifically, for the Phi–TACRED setting, we computed the Cohen's kappa coefficient to quantify inter-annotator agreement. The resulting kappa value was 0.693, indicating substantial agreement. This shows that our rubric is well-defined and practical. Therefore, we consistently adopted this rubric across all subsequent evaluations, including different stages and experimental settings.

### A.3  HUMAN EVALUATION RESULTS

To further evaluate explanation quality, we conducted human evaluation across three training stages: (i) vanilla LLMs with the COGRE framework, (ii) after RL training with the accuracy reward, and (iii) after RL training with the HIT@DICT reward.

We selected Qwen2.5-14B-Instruct and Phi-4 as base models and evaluated them on two datasets: one-shot TACRED and NYT29. For each LLM–dataset–stage combination, we sampled 40 explanations, with 10 explanations drawn from each of the four categories: true positives, true negatives, false positives, and false negatives. We consistently adopted the rubric we defined in the Appendix A.2. Therefore, for each category, the maximum score is 30 points.

We provide the results of human evaluation in Table 4. The results show that the models trained with the HIT@DICT reward consistently outperform both untrained models and accuracy-reward–trained models. For example, in the Phi–TACRED setting, the number of correct explanations in the no_yes and yes_no categories increases substantially after applying the HIT@DICT reward (from 24 to 29 and from 16 to 26, respectively). Similar improvements are observed in the Qwen–TACRED setting, particularly in the yes_no category (from 18 to 26).

| Setting / Category | Stages | | |
|---|---|---|---|
| | _untrained | _rl_ACC._reward | _rl_HIT@DICT_reward |
| ***Phi – TACRED*** | | | |
| no_yes | 24 | 26 | 29 |
| yes_no | 16 | 18 | 26 |
| yes_yes | 27 | 27 | 29 |
| no_no | 22 | 22 | 27 |
| ***Phi – NYT29*** | | | |
| no_yes | 18 | 25 | 24 |
| yes_no | 11 | 16 | 27 |
| yes_yes | 19 | 21 | 22 |
| no_no | 21 | 25 | 26 |
| ***Qwen – TACRED*** | | | |
| no_yes | 22 | 22 | 24 |
| yes_no | 18 | 18 | 26 |
| yes_yes | 26 | 26 | 29 |
| no_no | 19 | 21 | 25 |
| ***Qwen – NYT29*** | | | |
| no_yes | 19 | 22 | 20 |
| yes_no | 8 | 5 | 22 |
| yes_yes | 19 | 14 | 21 |
| no_no | 13 | 17 | 28 |

Table 4: Human evaluation results across four settings (Phi/Qwen × TACRED/NYT29). Each cell reports the score of a category in one combination under human evaluation. The categories are defined as follows: *no_yes*: the ground truth is No but the model prediction is Yes; *yes_no*: the ground truth is Yes but the model prediction is No; *yes_yes*: both the ground truth and the model prediction is Yes; *no_no*: both the ground truth and the model prediction is No;

## A.4 CASES COMPARISON OF PHI-4 ON TACRED

As discussed in Section 5.3, we examine the quality of explanations generated by LLMs across three stages: (1) vanilla LLM, (2) LLM trained with RL using only the accuracy reward, and (3) LLM trained with RL using both the accuracy reward and the HIT@DICT reward. We conducted evaluations on four model–dataset combinations, sampling and analyzing 40 instances for each. Here, we present examples from Phi-4 on the one-shot TACRED dataset. Specifically, we provide six illustrative cases: in five of them, the LLM trained with both accuracy and HIT@DICT rewards produces more concise summaries, and in all six cases, its explanations include relational keywords that align more closely with the gold relation labels. In the following examples, we highlight poor behaviors in red and the corresponding improvements in green.

| *Cases1 - Error Analysis from three Stages LLMs* |
|---|

▶ **Input:**
1. Federal regulators filed a civil fraud lawsuit Thursday against **Countrywide Financial** Corp co-founder **Angelo Mozilo** and two former associates.
2. It traveled to Rice University, where de Menil and his wife, **Dominique de Menil**, who later founded the **Menil Collection**, ran the art museum.

▶ **Gold Relation Labels and Ground Truth:**
1. org:founded_by
2. org:founded_by
3. Yes

▶ **Vanilla LLMs Output:**
Relation_Summarization_1: Federal regulators filed a civil fraud lawsuit against Angelo Mozilo, co-founder of Countrywide Financial Corp.
Relation_Summarization_2: Dominique de Menil founded the Menil Collection.
Understanding Process: - In Relation_Summarization_1, the relation is between a legal action (lawsuit) and Angelo Mozilo, who is associated with Countrywide Financial Corp as a co-founder. - In Relation_Summarization_2, the relation is between Dominique de Menil and the Menil Collection, where Dominique is the founder. The relations differ in nature: one involves a legal action against a co-founder, and the other involves founding an institution .

▶ **Trained LLMs with Acc. Output:**
Relation_Summarization_1:
Angelo Mozilo is a co-founder of Countrywide Financial Corp.
Relation_Summarization_2:
Dominique de Menil founded the Menil Collection.
Understanding Process:
- In Relation_Summarization_1, the relation is between a person (Angelo Mozilo) and a corporation (Countrywide Financial Corp), where the person is identified as a co-founder of the corporation.
- In Relation_Summarization_2, the relation is between a person (Dominique de Menil) and a collection (Menil Collection), where the person is identified as the founder of the collection.
- Both summaries describe a person's role in an entity, but the nature of the entities differs: one is a corporation , and the other is a collection .

▶ **Trained LLMs with HIT@DICT Output:**
Relation_Summarization_1: Angelo Mozilo is a co-founder of Countrywide Financial Corp.
Relation_Summarization_2: Dominique de Menil is the founder of Menil Collection.
Understanding Process:
In Relation_Summarization_1, the relation is organizational, indicating that Angelo Mozilo is a co-founder of Countrywide Financial Corp.
In Relation_Summarization_2, the relation is organizational, indicating that Dominique de Menil is the founder of Menil Collection.
In both cases, the relation is the same: a person is the founder of a company .

| *Cases2 - Error Analysis from three Stages LLMs* |
|---|

► **Input:**

1. The Honduran interim government would not be recognized by the international community, **Jacinto Suarez**, Nicaraguan deputy to the **Central American Parliament** ( PARLACEN ) said Monday.

2. SEC's McKown to Step Down as Chief Counsel for Enforcement Her replacement probably will inherit pending high-profile cases, including a fraud lawsuit against Goldman Sachs and an insider-trading case against **Angelo Mozilo**, the former chief executive officer of **Countrywide Financial** Corp.

► **Gold Relation Labels and Ground Truth:**

1. org:top_members/employees
2. org:top_members/employees
3. Yes

► **Vanilla LLMs Output:**

Relation_Summarization_1: Jacinto Suarez is a `Nicaraguan deputy` to the Central American Parliament.

Relation_Summarization_2: Angelo Mozilo is the former chief executive officer of Countrywide Financial Corp.

Understanding Process:

Relation_Summarization_1 describes Jacinto Suarez's role as a deputy to the Central American Parliament, indicating a membership or representative relationship.

Relation_Summarization_2 describes Angelo Mozilo's role as the former CEO of Countrywide Financial Corp, indicating a leadership or executive position within the company.

The nature of the relationships differs: one is a `representative role within a parliamentary body` ,

and the other is an `executive role within a corporation` .

Conclusion: No.

► **Trained LLMs with Acc. Output:**

Relation_Summarization_1: Jacinto Suarez is a `Nicaraguan deputy` to the Central American Parliament.

Relation_Summarization_2: Angelo Mozilo is the former chief executive officer of Countrywide Financial Corp.

Understanding Process:

Relation_Summarization_1 identifies Jacinto Suarez as holding a position (deputy) within the Central American Parliament.

Relation_Summarization_2 identifies Angelo Mozilo as having held a position (chief executive officer) within Countrywide Financial Corp.

Both summaries describe `a person and their role or position within an organization.`

The nature of the roles is similar in that both involve being part of an organization, but the specific roles differ (deputy vs. CEO).

Conclusion: No

► **Trained LLMs with HIT@DICT Output:**

Relation_Summarization_1: Jacinto Suarez is a `deputy` to the Central American Parliament.

Relation_Summarization_2: Angelo Mozilo is the former chief executive officer of Countrywide Financial Corp.

Understanding Process:

In Relation_Summarization_1, the relation is organizational, indicating that Jacinto Suarez holds the position of deputy to the Central American Parliament.

In Relation_Summarization_2, the relation is occupational, indicating that Angelo Mozilo is the former CEO of Countrywide Financial Corp.

In both cases, the relation is the same: a `person` is `employed` in a specific role within a

`company` / `organization` .

Yes

| Cases3 - Error Analysis from three Stages LLMs |
| --- |

► **Input:**

1. The religious re-awakening culminated in the 1998 visit by Catholic Pope John Paul II, but **Dworin** recalls with pride that the **Cuban** leader did not ignore the Jews.

2. In Port-au-Prince, **Laura Silsby**, the leader of the 10 **American** missionaries arrested at the Dominican border for trying to smuggle out a busload of children, appeared in court Monday to answer kidnapping charges.

► **Gold Relation Labels and Ground Truth:**

1. per:origin
2. per:origin
3. Yes

► **Vanilla LLMs Output:**

Relation_Summarization_1: Dworin recalls the Cuban leader acknowledging the Jews.

Relation_Summarization_2: Laura Silsby is the leader of the American missionaries.

Understanding Process:

1. Relation_Summarization_1 focuses on Dworin's recollection of the Cuban leader's acknowledgment of Jews, indicating a historical or cultural recognition .

2. Relation_Summarization_2 describes Laura Silsby's role as the leader of a group of American missionaries, indicating a leadership or organizational relationship.

The relations are not similar: the first is about historical acknowledgment, while the second is about leadership within a group.

► **Trained LLMs with Acc. Output:**

Relation_Summarization_1: Dworin recalls the Cuban leader acknowledging Jews.

Relation_Summarization_2: Laura Silsby is the leader of American missionaries.

Understanding Process:

Relation_Summarization_1 highlights a historical event where Dworin recalls a Cuban leader's acknowledgment of Jews, indicating a recognition or interaction.

Relation_Summarization_2 describes Laura Silsby's role as the leader of a group of American missionaries, indicating a leadership or organizational relationship.

The keywords in Relation_Summarization_1 ("recalls," "acknowledging") suggest a memory or recognition of an event involving a Cuban leader.

The keywords in Relation_Summarization_2 ("leader," "American missionaries") suggest a leadership role within a group.

The nature of the relationships is different: one is about historical acknowledgment , and the other is about leadership .

► **Trained LLMs with HIT@DICT Output:**

Relation_Summarization_1: Dworin is Cuban.

Relation_Summarization_2: Laura Silsby is American.

Understanding Process:

In Relation_Summarization_1, the relation is national identity , indicating that Dworin is Cuban.

In Relation_Summarization_2, the relation is identity , indicating that Laura Silsby is American.

In both cases, the relation is the same: A person is originally from a country.

| Cases4 - Error Analysis from three Stages LLMs |
| --- |

▶ **Input:**
1. This news comes from **Karr Ingham**, an economist who created the **Texas Petro Index** ( TPI ), which is a service of the Texas Alliance of Energy Producers.
2. **Ahmed Rashid**, a Pakistani journalist, is the author of "Descent into Chaos: The United States and the **Failure of Nation Building** in Afghanistan, Pakistan and Central Asia."

▶ **Gold Relation Labels and Ground Truth:**
1. org:founded_by
2. no_relation
3. No

▶ **Vanilla LLMs Output:**
Relation_Summarization_1: Karr Ingham created the Texas Petro Index.
Relation_Summarization_2: Ahmed Rashid is the author of "Failure of Nation Building."
Understanding Process:
Relation_Summarization_1 identifies Karr Ingham as the creator of the Texas Petro Index.
Relation_Summarization_2 identifies Ahmed Rashid as the author of "Failure of Nation Building."
The key relation in both summaries is one of creation or authorship.
In both cases, the first  subject  (Karr Ingham and Ahmed Rashid) is responsible for
 creating or authoring  the second  subject  (Texas Petro Index and "Failure of Nation Building").
Conclusion: Yes.

▶ **Trained LLMs with Acc. Output:**
Relation_Summarization_1: Karr Ingham created the Texas Petro Index.
Relation_Summarization_2: Ahmed Rashid is the author of "Failure of Nation Building."
Understanding Process:
Relation_Summarization_1 highlights a  creator-creation  relationship where Karr Ingham is the creator of the Texas Petro Index.
Relation_Summarization_2 highlights an  author-work  relationship where Ahmed Rashid is the author of "Failure of Nation Building."
Both summaries describe a relationship where one  entity  (Karr Ingham/Ahmed Rashid) is responsible for  producing or creating  another  entity  (Texas Petro Index/"Failure of Nation Building").
Conclusion: Yes.

▶ **Trained LLMs with HIT@DICT Output:**
Relation_Summarization_1: Karr Ingham is the creator of the Texas Petro Index.
Relation_Summarization_2: Ahmed Rashid is the author of "Failure of Nation Building."
Understanding Process:
In Relation_Summarization_1, the relation is  organizational , indicating that Karr Ingham is the person who created the Texas Petro Index.
In Relation_Summarization_2, the relation is  authorship , indicating that Ahmed Rashid is the author of a specific book.
The nature of these relations is different: one is about creation of an index, and the other is about authorship of a book.
No

| Cases5 - Error Analysis from three Stages LLMs |
|---|

▶ **Input:**

1. ( 1:32 ) R. " William Kunstler : Disturbing the Universe " – Filmmakers **Emily** and **Sarah Kunstler** explore the life of their father, the late radical civil rights lawyer who fought with Martin Luther King, Jr. and represented the famed "Chicago 8" activists who protested the Vietnam War.

2. **He** is also survived by his parents and a sister, Karen Lange, of Washington, and a brother, **Adam Lange**, of St Louis.

▶ **Gold Relation Labels and Ground Truth:**

1. per:siblings
2. per:siblings
3. Yes

▶ **Vanilla LLMs Output:**

Relation_Summarization_1: Sarah Kunstler and Emily are filmmakers exploring the life of their father.

Relation_Summarization_2: He is survived by his brother, Adam Lange.

Understanding Process:

- In Relation_Summarization_1, the relation is between Sarah Kunstler and Emily, who are collaborators in a filmmaking project.

- In Relation_Summarization_2, the relation is between "He" and Adam Lange, who are siblings.

The keywords in Relation_Summarization_1 ("filmmakers," "exploring") indicate a professional and collaborative relationship.

In Relation_Summarization_2 ("survived by," "brother"), the relationship is familial.

Conclusion: No

▶ **Trained LLMs with Acc. Output:**

Relation_Summarization_1: Sarah Kunstler and Emily are filmmakers exploring the life of their father.

Relation_Summarization_2: He and Adam Lange are siblings.

Understanding Process:

Relation_Summarization_1 identifies Sarah Kunstler and Emily as collaborators in a filmmaking project about their father.

Relation_Summarization_2 identifies a familial relationship between "He" and Adam Lange as siblings.

The keywords in Relation_Summarization_1 are "filmmakers" and "exploring," indicating a professional and collaborative relationship.

The keywords in Relation_Summarization_2 are "survived by" and "brother," indicating a familial relationship.

The nature of the relationships in both summaries is different: one is professional/collaborative, and the other is familial.

No

▶ **Trained LLMs with HIT@DICT Output:**

Relation_Summarization_1: Sarah Kunstler is Emily's.

Relation_Summarization_2: He is Adam Lange's brother.

Understanding Process:

In Relation_Summarization_1, the relation is familial, indicating that Sarah Kunstler and Emily are sisters.

In Relation_Summarization_2, the relation is also familial, indicating that "he" is Adam Lange's brother.

In both cases, the relation is the same: they are siblings.

Yes

| *Cases6 - Error Analysis from three Stages LLMs* |
|---|

▶ **Input:**
1.**China**'s National Development and Reform Commission (**NDRC**) on July 1 released a tape recording as evidence of the traders' mung bean price manipulation.
2. A US recession is likely to affect the industry, but strong growth in key Asian economies such as China and South Korea will drive regional travel demand, the **Pacific Asia Travel Association** (**PATA**) said.

▶ **Gold Relation Labels and Ground Truth:**
1. org:member_of
2. no_relation
3. No

▶ **Vanilla LLMs Output:**
Relation_Summarization_1:
NDRC is an abbreviation for China's National Development and Reform Commission.
Relation_Summarization_2: PATA is an abbreviation for the Pacific Asia Travel Association.
The relations in both summaries involve an abbreviation: NDRC is an abbreviation for China's National Development and Reform Commission, and PATA is an abbreviation for the Pacific Asia Travel Association.
Both summaries describe the relationship as one of abbreviation .
Yes.

▶ **Trained LLMs with Acc. Output:**
Relation_Summarization_1:
NDRC is an abbreviation for China's National Development and Reform Commission.
Relation_Summarization_2: PATA is an abbreviation for the Pacific Asia Travel Association.
Understanding Process: - In Relation_Summarization_1, the relation is that NDRC is an abbreviation for a specific organization within China. - In Relation_Summarization_2, the relation is that PATA is an abbreviation for a specific organization. - Both summaries describe an abbreviation relationship between an acronym and its full organizational name.
Conclusion: Yes.

▶ **Trained LLMs with HIT@DICT Output:**
Relation_Summarization_1: NDRC is China's National Development and Reform Commission.
Relation_Summarization_2: Pacific Asia Travel Association is abbreviated as PATA.
Understanding Process: In Relation_Summarization_1, the relation is organizational, indicating that NDRC is an organization within China. In Relation_Summarization_2, the relation is linguistic, indicating that PATA is an abbreviation for Pacific Asia Travel Association. The nature of these relations is different: one is about organizational identity within a country, and the other is about nomenclature.
No

A.5 STATISTICS OF THE SAMPLED TRAINING SET

To ensure fairness and representativeness in one-shot relation extraction evaluation, we construct sampled training sets for both one-shot NYT29 and TACRED that preserve the distributional properties of the original datasets. The algorithm of sampling training data is shown in Algorithm 2.

---

**Algorithm 2** Sampling Procedure for Training Data

---

**Require:** Original dataset $\mathcal{D}$; sampling quotas $Q$; maximum positives per label $K$
**Ensure:** Sampled dataset $\mathcal{D}'$
 1: Split $\mathcal{D}$ into subsets:
   - $\mathcal{D}_{r,r}$: $ss\_relation = ts\_relation$                                  ▷ Positive pairs
   - $\mathcal{D}_{r,no}$: $ts\_relation = $ no_relation                        ▷ One relation + no_relation
   - $\mathcal{D}_{r,r'}$: $ss\_relation \neq ts\_relation$, $ts\_relation \neq $ no_relation        ▷ Different relations
 2: $\mathcal{D}' \leftarrow \emptyset$
 3: From $\mathcal{D}_{r,r}$: if a relation has more than $K$ pairs, randomly down-sample to $K$; otherwise keep all. Add to $\mathcal{D}'$.
 4: From $\mathcal{D}_{r,no}$: group by $ss\_relation$. For each relation $r$, sample $Q[r]$ pairs (or all if fewer available). Add to $\mathcal{D}'$.
 5: From $\mathcal{D}_{r,r'}$: randomly sample $2,583$ pairs, approximately preserving label distribution. Add to $\mathcal{D}'$.
 6: Shuffle $\mathcal{D}'$.
 7: Report statistics: $|\mathcal{D}'|$, #positives, #negatives, and ratio.
 8: **return** $\mathcal{D}'$

---

Tables 5 and 6 report the ratio of positives to negatives in the training partition of one-shot NYT29 and TACRED, respectively. In these tables, we compare the ratio of positives to negatives, and the proportion of one of relation labels-no_relation before and after sampling. For NYT29, it can be seen that the ratio of positive items $(r, r)$, negative items with no_relation, $(r, \text{no\_relation})$, and negative items without no_relation, $(r, r')$ in the sampled set aligns well with the original dataset. A similar pattern holds for TACRED, where the sampled data maintains the same balance across positive and negative categories.

| | Original | Sampled |
|---|---|---|
| Positive items $(r, r)$ | 71376 | 2670 |
| Negative items with no_relation $(r, \text{no\_relation})$ | 571560 | 9660 |
| Negative items without no_relation $(r, r')$ | 285504 | 7670 |

Table 5: Ratio of positives and negatives on the original training partition of one-shot NYT29 and our sampled version. *Positive items* $(r, r)$: pairs where both sentences express the same relation $r$. *Negative items with no_relation* $(r, \text{no\_relation})$: pairs where one sentence expresses a relation $r$ and the other is labeled as no_relation. *Negative items without no_relation* $(r, r')$: pairs where the two sentences express different relations $r$ and $r'$, neither being no_relation.

| | Original | Sampled |
|---|---|---|
| Positive items $(r, r)$ | 7170 | 2583 |
| Negative items with no_relation $(r, no\_relation)$ | 732075 | 14834 |
| Negative items without no_relation $(r, r')$ | 28680 | 2583 |

Table 6: Ratio of positives and negatives on the original training partition of one-shot TACRED and our sampled version. *Positive items* $(r, r)$: pairs where both sentences express the same relation $r$. *Negative items with no_relation* $(r, \text{no\_relation})$: pairs where one sentence expresses a relation $r$ and the other is labeled as no_relation. *Negative items without no_relation* $(r, r')$: pairs where the two sentences express different relations $r$ and $r'$, neither being no_relation.

| Relation | Original | Sampled |
|---|---|---|
| /business/company/founders | 52,201 | 1,634 |
| /business/company/place_founded | 51,408 | 1,571 |
| /business/person/company | 66,753 | 2,226 |
| /film/film_location/featured_in_films | 50,425 | 1,590 |
| /location/country/capital | 70,537 | 2,351 |
| /location/location/contains | 135,244 | 4,934 |
| /location/us_county/county_seat | 50,394 | 1,572 |
| /location/us_state/capital | 51,938 | 1,681 |
| /people/deceased_person/place_of_burial | 49,984 | 1,545 |
| /people/ethnicity/geographic_distribution | 51,557 | 1,608 |
| /people/person/children | 51,332 | 1,553 |
| /people/person/ethnicity | 50,625 | 1,566 |
| /people/person/nationality | 74,607 | 2,511 |
| /people/person/place_lived | 71,195 | 2,437 |
| /people/place_of_interment/interred_here | 50,240 | 1,561 |
| no_relation | 571,560 | 9,660 |

Table 7: Distribution of relation labels in the one-shot NYT29 training partition (original vs. sampled). Each row corresponds to a relation type, where the *Original* column reports the number of instances in the full dataset and the *Sampled* column reports the number of instances included in our one-shot sampled version. The relation `no_relation` indicates sentence pairs that do not express any annotated relation.

In addition, Tables 7 and 8 further analyze the distribution of all the relation labels in the original dataset and the sampled dataset, respectively. Since we sampled the training data strictly according to the original distribution of relation labels, the sampled training datasets have a similar label distribution to the original ones. These statistics show that our sampled training datasets faithfully reflect the statistical properties of the original datasets, thereby avoiding biases introduced by over- or under-sampling specific relations.

| Relation | Original | Sampled |
|---|---|---|
| org:alternate_names | 31,532 | 1,156 |
| org:city_of_headquarters | 31,016 | 997 |
| org:dissolved | 30,011 | 912 |
| org:members | 30,038 | 968 |
| org:number_of_employees/members | 30,236 | 929 |
| org:political/religious_affiliation | 30,633 | 938 |
| org:shareholders | 30,288 | 945 |
| org:stateorprovince_of_headquarters | 30,702 | 1,011 |
| org:subsidiaries | 30,279 | 988 |
| org:website | 30,307 | 946 |
| per:cause_of_death | 30,388 | 947 |
| per:charges | 29,975 | 930 |
| per:cities_of_residence | 30,833 | 1,003 |
| per:city_of_birth | 30,118 | 912 |
| per:countries_of_residence | 30,870 | 1,035 |
| per:country_of_birth | 29,958 | 907 |
| per:country_of_death | 29,833 | 903 |
| per:date_of_death | 30,644 | 940 |
| per:employee_of | 32,941 | 1,383 |
| per:other_family | 30,106 | 976 |
| per:parents | 30,290 | 959 |
| per:religion | 30,145 | 935 |
| per:spouse | 30,576 | 967 |
| per:stateorprovince_of_birth | 30,293 | 908 |
| per:title | 35,913 | 1,671 |
| no_relation | 732,075 | 14,834 |

Table 8: Distribution of relation labels in the one-shot TACRED training partition (original vs. sampled). Each row corresponds to a relation type, where the *Original* column reports the number of instances in the full dataset and the *Sampled* column reports the number of instances included in our one-shot sampled version. The relation `no_relation` indicates sentence pairs that do not express any annotated relation.

## A.6 STATISTICS OF THE SAMPLED TESTING SET

For the testing partition, we adopt a different sampling strategy from the training data, using a random sampling approach. The resulting sampled testing set preserves the same distributional properties as the original dataset. Specifically, (1) the ratio of positive to negative instances remains consistent, (2) the distribution of relation labels is well aligned, and (3) the proportion of no_relation instances is maintained. These results confirm that our random sampling strategy produces a representative testing partition that faithfully reflects the characteristics of the original testing dataset.

| Category | Original | Sampled |
|---|---|---|
| Positive items $(r, r)$ | 7,055 | 704 |
| Negative items with no_relation $(r, no\_relation)$ | 114,725 | 11,480 |
| Negative items without no_relation $(r, r')$ | 28,220 | 2,816 |

Table 9: Ratio of positives and negatives on the original testing partition of one-shot NYT29 and our sampled version. *Positive items* $(r, r)$: pairs where both sentences express the same relation $r$. *Negative items with no_relation* $(r, \text{no\_relation})$: pairs where one sentence expresses a relation $r$ and the other is labeled as no_relation. *Negative items without no_relation* $(r, r')$: pairs where the two sentences express different relations $r$ and $r'$, neither being no_relation.

| Category | Original | Sampled |
|---|---|---|
| Positive items $(r, r)$ | 772 | 82 |
| Negative items with no_relation $(r, no\_relation)$ | 146,140 | 14,590 |
| Negative items without no_relation $(r, r')$ | 3,088 | 328 |

Table 10: Ratio of positives and negatives on the original testing partition of one-shot TACRED and our sampled version. *Positive items* $(r, r)$: pairs where both sentences express the same relation $r$. *Negative items with no_relation* $(r, \text{no\_relation})$: pairs where one sentence expresses a relation $r$ and the other is labeled as no_relation. *Negative items without no_relation* $(r, r')$: pairs where the two sentences express different relations $r$ and $r'$, neither being no_relation.

| Relation | Original | Sampled |
|---|---|---|
| /business/company/major_shareholders | 30,510 | 3,055 |
| /location/administrative_division/country | 40,690 | 4,130 |
| /location/country/administrative_divisions | 41,160 | 4,040 |
| /location/neighborhood/neighborhood_of | 39,520 | 3,960 |
| /people/deceased_person/place_of_death | 33,395 | 3,335 |
| no_relation | 114,725 | 11,480 |

Table 11: Distribution of relation labels in the one-shot NYT29 testing partition (original vs. sampled). Each row corresponds to a relation type, where the *Original* column reports the number of instances in the full dataset and the *Sampled* column reports the number of instances included in our one-shot sampled version. The relation no_relation indicates sentence pairs that do not express any annotated relation.

| Relation | Original | Sampled |
|---|---|---|
| no_relation | 146,140 | 14,590 |
| org:founded_by | 15,442 | 1,611 |
| org:member_of | 15,246 | 1,557 |
| org:top_members/employees | 16,431 | 1,648 |
| per:children | 14,977 | 1,413 |
| per:city_of_death | 14,944 | 1,535 |
| per:date_of_birth | 15,127 | 1,528 |
| per:origin | 15,513 | 1,582 |
| per:schools_attended | 15,148 | 1,449 |
| per:siblings | 15,333 | 1,519 |
| per:stateorprovinces_of_residence | 15,699 | 1,568 |

Table 12: Distribution of relation labels in the one-shot TACRED testing partition (original vs. sampled). Each row corresponds to a relation type, where the *Original* column reports the number of instances in the full dataset and the *Sampled* column reports the number of instances included in our one-shot sampled version. The relation `no_relation` indicates sentence pairs that do not express any annotated relation.

## A.7 PROMPT FOR COGRE FRAMEWORK

---

**COGRE Prompt**

You are given two sentences. Follow the three steps below to determine whether they express a similar relation.

—

Summarization: Focus on the main parts between subjects and objects in the sentences.
Summarization examples:

Summarize the relations between "Malcolm Peeler" and "Pangburn" in "Dr. Malcolm Peeler , grew in Pangburn, has continued the family tradition of practicing medicine in Jonesboro .".
Summarization: Malcolm Peeler came from Pangburn.

Summarize the relations between "Oceania" and "PECC" in "Oceania and the Western Hemisphere within the PECC region , as surplus food producers and exporters , confront unique consumer issues , such as lower food expenditure and higher caloric intake compared to Asia .".
Summarization: Oceania within region PECC.

Summarize the relations between "Global Climate Research Institute" and "GCRI" in "Climate change challenges remain a key concern at the annual summit. The outlook is concerning, according to the Global Climate Research Institute ( GCRI ), which coordinates the event each year.".
Summarization: Global Climate Research Institute is abbreviated as GCRI.

Summarize the relations between "Panasonic Corp" and "Tesla Inc" in "Tesla Inc. is a wholly-owned subsidiary of Panasonic Corp, focusing on energy storage solutions.".
Summarization: Panasonic Corp is a subsidiary of Tesla Inc.

—

Step 1: summarize the relations between "{support_sentence_subject}" and "{support_sentence_object}" in "{support_sentence}".
Label your result as: Relation_Summarization_1.

Step 2: summarize the relations between "{test_sentence_subject}" and "{test_sentence_object}" in "{test_sentence}".
Label your result as: Relation_Summarization_2.

Step 3: are the relations between "{support_sentence_subject}" and "{support_sentence_object}" in Relation_Summarization_1 and between "{test_sentence_subject}" and "{test_sentence_object}" in Relation_Summarization_2 similar?
Focus on the keywords in the Relation_Summarization_1 an Relation_Summarization_2 that convey relations.

Generate the understanding process, followed by Yes or No in a separate line.

---

## A.8 PROMPT FOR RELATION KEYWORD EXTRACTION

**Keyword Extraction Prompt for GPT-4o**

Relation: {relation}
Please extract the words or phrases that indicate trigger words or relation summaries from the following answers; the relation is {relation}.
Output a string list contain all the words.
output_case_1:
{content_1}
support_sentence: {support_sentence_1}
test_sentence: {test_sentence_1}

output_case_2:
{content_2}
support_sentence: {support_sentence_2}
test_sentence: {test_sentence_2}

output_case_3:
{content_3}
support_sentence: {support_sentence_3}
test_sentence: {test_sentence_3}

output_case_4:
{content_4}
support_sentence: {support_sentence_4}
test_sentence: {test_sentence_4}

output_case_5:
{content_5}
support_sentence: {support_sentence_5}
test_sentence: {test_sentence_5}

## A.9 PROMPT FOR BASELINES

---

**Direct-Answer Prompt**

Are the relations between "{support_sentence_subject}" and "{support_sentence_object}" in "{support_sentence}" and between "{test_sentence_subject}" and "{test_sentence_object}" in {test_sentence} similar? Directly answer Yes or No in a separate line.

—

IMPORTANT: must answer with just Yes or No.

---

**Simple-Reasoning Prompt**

Are the relations between "{support_sentence_subject}" and "{support_sentence_object}" in "{support_sentence}" and between "{test_sentence_subject}" and "{test_sentence_object}" in {test_sentence} similar?
Generate the understanding process, followed by Yes or No in a separate line.

---

## A.10 PROMPTS FOR SUMASK

---

**one-prompt SUMASK Prompt**

You are given two sentences. Follow the three steps below to determine whether they express a similar relation.

—

Summarization examples:

Summarize the relations between "Malcolm Peeler" and "Pangburn" in "Dr. Malcolm Peeler , grew in Pangburn, has continued the family tradition of practicing medicine in Jonesboro .".
Summarization: Malcolm Peeler came from Pangburn.

Summarize the relations between "Oceania" and "PECC" in "Oceania and the Western Hemisphere within the PECC region , as surplus food producers and exporters , confront unique consumer issues , such as lower food expenditure and higher caloric intake compared to Asia .".
Summarization: Oceania within region PECC.

Summarize the relations between "Global Climate Research Institute" and "GCRI" in "Climate change challenges remain a key concern at the annual summit. The outlook is concerning, according to the Global Climate Research Institute ( GCRI ), which coordinates the event each year.".
Summarization: Global Climate Research Institute is abbreviated as GCRI.

Summarize the relations between "Panasonic Corp" and "Tesla Inc" in "Tesla Inc. is a wholly-owned subsidiary of Panasonic Corp, focusing on energy storage solutions.".
Summarization: Panasonic Corp is a subsidiary of Tesla Inc.

—

Step 1: summarize the relations between "{support_sentence_subject}" and "{support_sentence_object}" in "{support_sentence}".
Label your result as: Relation_Summarization_1.

Step 2: summarize the relations between "{test_sentence_subject}" and "{test_sentence_object}" in "{test_sentence}".
Label your result as: Relation_Summarization_2.

Step 3: generate a question as: are the relations between "{support_sentence_subject}" and "{support_sentence_object}" in Relation_Summarization_1 and between "{test_sentence_subject}" and "{test_sentence_object}" in Relation_Summarization_2 similar?

Step 4: directly answer the question with Yes or No in a separate line.

---

## A.11 Prompt for Ablation experiments

---

**CoGRE- *w/o* chunking Prompt**

You are given two sentences. Follow the three steps below to determine whether they express a similar relation.
Step1: are the relations between "{paraphrased_sentence_subject}" and
"{paraphrased_sentence_object}" in {paraphrased_sentence} and between "{test_sentence_subject}" and "{test_sentence_object}" in {test_sentence} similar? Focus on the keywords in the {paraphrased_sentence} an {test_sentence} that convey relations.
Generate the understanding process, followed by Yes or No in a separate line.

---

**CoGRE- *w/o* reasoning Prompt**

You are given two sentences. Follow the three steps below to determine whether they express a similar relation.
—
Summarization examples:

Summarize the relations between "Malcolm Peeler" and "Pangburn" in "Dr. Malcolm Peeler , grew in Pangburn, has continued the family tradition of practicing medicine in Jonesboro .".
Summarization: Malcolm Peeler came from Pangburn.

Summarize the relations between "Oceania" and "PECC" in "Oceania and the Western Hemisphere within the PECC region , as surplus food producers and exporters , confront unique consumer issues , such as lower food expenditure and higher caloric intake compared to Asia .".
Summarization: Oceania within region PECC.

Summarize the relations between "Global Climate Research Institute" and "GCRI" in "Climate change challenges remain a key concern at the annual summit. The outlook is concerning, according to the Global Climate Research Institute ( GCRI ), which coordinates the event each year.". Summarization: Global Climate Research Institute is abbreviated as GCRI.

Summarize the relations between "Panasonic Corp" and "Tesla Inc" in "Tesla Inc. is a wholly-owned subsidiary of Panasonic Corp, focusing on energy storage solutions.".
Summarization: Panasonic Corp is a subsidiary of Tesla Inc.
—
Step 1: summarize the relations between "{support_sentence_subject}" and
"{support_sentence_object}" in "{support_sentence}".
Label your result as: Relation_Summarization_1.

Step 2: summarize the relations between "{test_sentence_subject}" and "{test_sentence_object}" in "{test_sentence}".
Label your result as: Relation_Summarization_2.

Step 3: generate a question as: are the relations between "{support_sentence_subject}" and "{support_sentence_object}" in Relation_Summarization_1 and between "{test_sentence_subject}" and "test_sentence_object}" in Relation_Summarization_2 similar?

Step 4: Focus on the keywords in the Relation_Summarization_1 an Relation_Summarization_2 that convey relations, and directly answer the question with Yes or No in a separate line.

---

**COGRE- *w/o* keyword Prompt**

You are given two sentences. Follow the three steps below to determine whether they express a similar relation.
—
Summarization examples:

Summarize the relations between "Malcolm Peeler" and "Pangburn" in "Dr. Malcolm Peeler , grew in Pangburn, has continued the family tradition of practicing medicine in Jonesboro .".
Summarization: Malcolm Peeler came from Pangburn.

Summarize the relations between "Oceania" and "PECC" in "Oceania and the Western Hemisphere within the PECC region , as surplus food producers and exporters , confront unique consumer issues , such as lower food expenditure and higher caloric intake compared to Asia .".
Summarization: Oceania within region PECC.

Summarize the relations between "Global Climate Research Institute" and "GCRI" in "Climate change challenges remain a key concern at the annual summit. The outlook is concerning, according to the Global Climate Research Institute ( GCRI ), which coordinates the event each year.". Summarization: Global Climate Research Institute is abbreviated as GCRI.

Summarize the relations between "Panasonic Corp" and "Tesla Inc" in "Tesla Inc. is a wholly-owned subsidiary of Panasonic Corp, focusing on energy storage solutions.".
Summarization: Panasonic Corp is a subsidiary of Tesla Inc.
—
Step 1: summarize the relations between "{support_sentence_subject}" and "{support_sentence_object}" in "{support_sentence}".
Label your result as: Relation_Summarization_1.

Step 2: summarize the relations between "{test_sentence_subject}" and "{test_sentence_object}" in "{test_sentence}".
Label your result as: Relation_Summarization_2.

Step 3: generate a question as: are the relations between "{support_sentence_subject}" and "{support_sentence_object}" in Relation_Summarization_1 and between "{test_sentence_subject}" and "{test_sentence_object}" in Relation_Summarization_2 similar?
Generate the understanding process, followed by Yes or No in a separate line.

## A.12 INFERENCE RESULTS ACROSS MODEL FAMILIES AND SIZES ON ONE-SHOT RE TASK

In addition to Phi-4 and Qwen2.5-14B-Instruct, we further evaluate our COGRE reasoning method on models from four additional families of varying sizes. The results, shown in Table 13, reveal two key findings: (1) COGRE consistently outperforms prompting-based baselines across both model families and model sizes; and (2) models with fewer than 10B parameters perform poorly on the one-shot RE task.

| Technique | One-shot TACRED | | |
|---|---|---|---|
| | P | R | F1 |
| *Qwen2.5-7B-Instruct* | | | |
| Direct Matching | 22.58 | 9.21 | 13.08 |
| Random Reasoning | 3.42 | 43.42 | 6.35 |
| Cognitive-Structured RE (*our*) | 10.24 | 50.00 | 17.00 |
| *Qwen2.5-3B-Instruct* | | | |
| Direct Matching | 100.00 | 0.00 | 0.00 |
| Random Reasoning | 10.53 | 7.89 | 9.02 |
| Cognitive-Structured RE (*our*) | 4.69 | 36.84 | 8.32 |
| *Phi-4-mini-Instruct (4B)* | | | |
| Direct Matching | 0.88 | 28.95 | 1.70 |
| Random Reasoning | 1.65 | 52.63 | 3.21 |
| Cognitive-Structured RE (*our*) | 9.35 | 34.21 | 14.69 |
| *Llama-3.2-8B-Instruct* | | | |
| Direct Matching | 1.61 | 39.47 | 3.09 |
| Random Reasoning | 0.43 | 17.11 | 0.85 |
| Cognitive-Structured RE (*our*) | 0.72 | 27.63 | 1.40 |
| *Mistral-7B-2v* | | | |
| Direct Matching | 5.00 | 17.11 | 7.74 |
| Random Reasoning | 0.95 | 26.32 | 1.83 |
| Cognitive-Structured RE (*our*) | 5.14 | 25.00 | 8.52 |

Table 13: Performance comparison of different model families and sizes on the one-shot TACRED task. We evaluate our COGRE framework against prompting-based baselines (Direct Matching and Random Reasoning).

