# OpenReview forum: "Peeking inside the Black-Box: Reinforcement Learning for Explainable and Accurate Relation Extraction"
_ICLR.cc/2026/Conference — Submitted to ICLR 2026_

### Official Review · Reviewer_Pg7E · 2025-10-26

**Soundness:** 3
**Presentation:** 3
**Contribution:** 2
**Rating:** 4
**Confidence:** 3

**Summary:**

This paper proposes a novel framework called COGRE for relation extraction (RE) tasks, aiming to improve both accuracy and explainability. The framework combines cognitive-structured reasoning and reinforcement learning (RL) to optimize RE models. A key component of the method is the HIT@DICT reward, which uses a relationship keywords dictionary to evaluate the quality of explanations generated by the model. The authors claim that this approach improves RE performance, especially for one-shot learning, and provides better human-readable explanations. The framework is evaluated on TACRED and NYT29 datasets, showing improved F1 scores and human evaluation results compared to baselines.

**Strengths:**

The paper introduces a novel approach by combining cognitive-structured reasoning with reinforcement learning for relation extraction tasks. This integration allows the model to improve both the accuracy and explainability of its predictions, which is a significant contribution to the field of explainable AI. The experimental setup is well-designed, and the results, particularly in terms of F1 score and human evaluation, show that the proposed HIT@DICT reward improves the model’s ability to generate better explanations.

**Weaknesses:**

The HIT@DICT reward relies heavily on a predefined keywords dictionary, which restricts the model’s ability to generalize beyond the fixed set of keywords, potentially causing bias and limiting performance in more complex or novel relations. This reliance on keywords could also lead to overfitting to specific patterns, rather than allowing the model to capture the full complexity of relational semantics. Second, the lack of comparison with current mainstream large models (like GPT-4o，GPT-5，Gemini) is a major drawback, as it makes it difficult to assess how this approach stacks up against state-of-the-art methods. Without this comparison, it's unclear whether the proposed framework brings substantial improvement over existing models, which have demonstrated strong performance in similar tasks.

**Questions:**

Can the authors provide more details on how the keywords dictionary is built, and what the limitations of this approach might be in real-world applications? How would the model perform when faced with novel relations that do not have clear keyword representations in the dictionary?

Would it be possible to expand the experiments to include comparisons with mainstream LLMs? How does COGRE compare with these models in terms of both accuracy and explanation quality?

---

> ### Author Response · Authors · 2025-11-20
>
> Dear Reviewer,
>
> Thank you for your comments. We truly appreciate your thoughtful suggestions and guidance, which help us improve our work. In response to your review, we have made the following revisions:
>
> 1. Clarified that Hit@Dict does not show evidence of inducing keyword overfitting and demonstrated improved generalization to unseen relations. (Addressing Weakness 1)
> 2. Explained how the dictionary is constructed and addressed concerns about generalization to novel relations and potential pattern overfitting. (Addressing Question 1)
> 3. Reported our evaluation results of mainstream LLMs and explained why only open-source models were reported in paper. (Addressing Weakness 2 and Question 2)
>
> More details are provided below.

---

> ### Author Response · Authors · 2025-11-20
> **Response to Weakness 1 about Potential Overfitting and Generalization Issue**
>
> Our keyword dictionary does not show evidence of restricting the model’s ability to generalize beyond the fixed keyword set, nor do we observe indications of introducing bias, limiting performance, or causing pattern-level overfitting. Instead, it appears to help the model capture underlying labeling patterns in relation datasets. This is supported by several observations and empirical results:
>
> **(1) Improved Alignment on Unseen Relation Labels:** our Hit@Dict reward improves the model’s alignment with relation labels that are unseen during training. As discussed in the second paragraph of the *CogRE* section in 5.3 and in the fourth contribution of the Introduction, our human evaluation shows that models trained with Acc + Hit@Dict exhibit better alignment with gold relation labels compared to models trained with Acc-only. As noted in the *Benchmark* section (Section 4), the relation labels in the training and testing partitions are out-of-distribution, meaning the model never sees the test relations during training. This evidence indicates that our reward does not introduce bias or limit performance on more complex or novel relations. Instead, it enhances the model’s generalization ability in relation extraction. Importantly, the model is unlikely to be merely reproducing textual keywords; Rather, our analysis indicates that it captures regularities that correspond to relation-labeling patterns in the dataset.
>
> **(2) A Highly Adaptable Keyword-Dictionary Pipeline for novel Training dataset:**  our keyword-dictionary construction pipeline can be easily applied to novel training datasets. Because it automatically extracts keywords from the base model’s own high-quality explanations, it does not require human-crafted relation label descriptions. This makes the pipeline easy to adapt and quick to deploy on new or previously unseen datasets.
>
> **(3) Our Hit@Dict simply rewards the model to generate relation-relevant keywords that the model itself produces.** The reward doesn't exposes the model to relation labels through SFT supervision, nor does it provide any label information through prompting. Instead, the model learns through a trial-and-error process to “discover” effective strategies on its own. As a result, compared with other training paradigms, it is less prone to overfitting to a specific keyword set; rather, it encourages the model to infer the relation from the sentence instead of memorizing labels or imitating predefined keywords.
>
> **(4) No Pattern Overfitting Observed in Empirical Results:** our **empirical results** show no signs of pattern overfitting, and the model trained with Acc + Hit@Dict explains unseen relations better than the model trained with Acc-only. As demonstrated in the case studies in Appendix A.4, the explanations generated in the *Trained LLMs with Hit@Dict Output* cases are both reasonable and well aligned with the unseen relations in the test set. Across all 160 evaluated cases, we did not observe any behavior that indicates overfitting to specific patterns or keyword sets.

---

> ### Author Response · Authors · 2025-11-20
> **Response to Question 1 about Dictionary Construction and Generalization to Novel Relations**
>
> **(1) Dictionary Construction:** As shown in Algorithm 1 (the construction procedure), Figure 1 (pipeline shown in the upper-left), and Section 3.2 (lines 239–246, *“How to Construct a Relational Keywords Dictionary?”*), our paper provides a detailed description of how the dictionary is built. **In summary**, the relational keyword dictionary is constructed automatically, using only a small number of correctly predicted positive examples per relation (typically 1–5). A LLM extracts keywords from these examples, and we additionally perform simple lexical decomposition of the relation label name. The extracted terms are then cleaned through standard post-processing steps (tokenization, deduplication, and lemmatization). The resulting dictionary is lightweight, requires no human-crafted label descriptions, and is generated entirely from data and base-model outputs. This makes the dictionary construction pipeline easy to adapt and fast to apply to novel relation datasets. This directly contributes to solving one of the main challenges outlined in the Introduction, namely “(2) in some cases, the need for handcrafted training datasets that are expensive to annotate.”
>
> **(2) Potential Limitations and Generalization to Novel Relations:** The main real-world limitation you may be concerned about is the generalization ability of our method. Below, we address several specific sub-questions related to generalization that you may be considering.
>
> * **How would the model perform when encountering novel relations that do not have clear keyword representations in the dictionary?** We discuss this question under two scenarios:
>   * **Scenario 1: Novel relations appearing in a new training set.** Because our pipeline automatically extracts keywords from the base model’s own high-quality explanations, it does **not** require any human-crafted label descriptions. This makes the keyword-dictionary construction process easy to adapt and fast to apply to new training datasets. Whenever a new dataset introduces novel relations, we can simply rerun the pipeline to build a new dictionary without manually designing any relation descriptions for those new labels.
>   * **Scenario 2: Novel and unseen relations appearing in the testing set.** Our Hit@Dict reward improves the model’s alignment with relation labels that are *unseen* during training. As discussed in the second paragraph of the **CogRE** section in 5.3 and in the fourth contribution of the **Introduction**, our human evaluation shows that models trained with **Acc + Hit@Dict** align more closely with the gold relation labels in the test set compared to models trained with **Acc-only**. Moreover, as described in the **Benchmark** section (Section 4), the relation labels in the training and testing partitions are **out-of-distribution**, meaning the test relations never appear during training.
>
> * **Does our method cause models to overfit to specific patterns?**
>   * **From a functional perspective,** our Hit@Dict rewards only the relation-relevant keywords that the model itself generates, rather than providing any external label information. The reward does not expose the model to relation labels through SFT supervision, nor does it encode label semantics through prompting. Instead, the model learns through a trial-and-error process to discover effective strategies on its own. As a result, it does not overfit to a predefined keyword set but instead learns to infer the underlying relation from the sentence, rather than memorizing labels or imitating fixed patterns.
>   * **From empirical evidence,** we observe no signs of pattern overfitting. Models trained with Acc+Hit@Dict generate explanations that align better with unseen test relations than those trained with Acc-only. As shown in Appendix A.4, the explanations Hit@Dict cases are reasonable and well matched to the unseen relations, with no indication of overfitting to specific keyword patterns.

---

> ### Author Response · Authors · 2025-11-20
> **Response to Weakness 2 and Question2 about Mainstream Large Language Models**
>
> **As reported in Appendix A.12 (Table 13)**, we have already evaluated a wide range of models under our setting, **including 3B–7B language models** (Qwen2.5-3B/7B, Phi-4-4B, Llama-3.2-8B, Mistral-7B-2v) **as well as large mainstream LLMs** such as DeepSeek-v3, GPT-4o, and GPT-3.5-turbo. **However, we only report the first category (3B–7B open-source models) in our paper**. The cost of running mainstream LLM APIs is prohibitively high, which limits reproducibility and accessibility for the community. To ensure fairness and feasibility, we therefore only report the open-source models.
>
> **Now, we included all the results below.** The dataset is the dev partition of one-shot TACRED. We observe that models perform poorly under our setting, whether using *Random-Reasoning-then-Answer* or *Direct-Answer*. We further find that generating reasoning often makes models over-confident, leading to more False Positive predictions. These results and findings demonstrate that the realistic Few-Shot Relation Extraction setting is still challenging even for cutting-edge LLMs. Therefore our proposed framework brings substantial improvement and achieves strong performance where existing LLMs fail.
>
> | Model                    | Method                       | Setting | Precision | Recall | F1     |
> | ------------------------ | ---------------------------- | ------- | --------- | ------ | ------ |
> | Before RL-Qwen2.5-14B   | CogRE-Reasoning          | ICL     | 35.29%    | 39.47% | 37.27% |
> | Deepseek-v3              | Direct-Answer                | ICL     | 35.62%    | 37.14% | 36.36% |
> | Deepseek-v3              | Random-Reasoning-then-Answer | ICL     | 38.46%    | 21.43% | 27.52% |
> | GPT-4o                   | Random-Reasoning-then-Answer | ICL     | 20.00%    | 23%    | 21.33% |
> | GPT-3.5-turbo            | Random-Reasoning-then-Answer | ICL     | 16.23%    | 36%    | 22.32% |
> | Qwen2.5-7B-Instruct      | Direct-Answer                | ICL     | 22.58%    | 9.21%  | 13.08% |
> | Qwen2.5-7B-Instruct      | Random-Reasoning-then-Answer | ICL     | 3.42%     | 43.42% | 6.35%  |
> | Qwen2.5-3B-Instruct      | Direct-Answer                | ICL     | 100%      | 0%     | 0%     |
> | Qwen2.5-3B-Instruct      | Random-Reasoning-then-Answer | ICL     | 10.53%    | 7.89%  | 9.02%  |
> | Phi-4-mini-Instruct (4B) | Direct-Answer                | ICL     | 0.88%     | 28.95% | 1.70%  |
> | Phi-4-mini-Instruct (4B) | Random-Reasoning-then-Answer | ICL     | 1.65%     | 52.63% | 3.21%  |
> | Llama-3.2-8B-Instruct    | Direct-Answer                | ICL     | 1.61%     | 39.47% | 3.09%  |
> | Llama-3.2-8B-Instruct    | Random-Reasoning-then-Answer | ICL     | 0.43%     | 17.11% | 0.85%  |
> | Mistral-7B-2v            | Direct-Answer                | ICL     | 5%        | 17.11% | 7.74%  |
> | Mistral-7B-2v            | Random-Reasoning-then-Answer | ICL     | 0.95%     | 26.32% | 1.83%  |

---

> > ### Comment · Reviewer_Pg7E · 2025-11-25
> >
> > The rebuttal clarified some points of confusion. I have raised the contribution score, but I maintain my original assessment of the overall score.

---

> > > ### Author Response · Authors · 2025-11-25
> > >
> > > Dear Reviewer Pg7E,
> > >
> > > Thank you very much for your thoughtful engagement with our work.
> > >
> > > To make the best use of the remaining discussion time, we would like to kindly check whether there are any concerns that our responses have not yet addressed. Please let us know if any part of our response is unclear or if there are any remaining issues we can clarify. We would be more than happy to provide any further details or adjustments you might need.
> > >
> > > Best wishes,
> > > Authors

---

### Official Review · Reviewer_GtZe · 2025-10-31

**Soundness:** 2
**Presentation:** 3
**Contribution:** 2
**Rating:** 4
**Confidence:** 4

**Summary:**

This paper introduces COGRE, a Relation Matching framework improving accuracy and explainability using a cognitive-inspired, three-step reasoning prompt (chunking, anchoring, reasoning). This reasoning is then optimized via RL using a novel HIT@DICT reward. This reward combines task accuracy with an explanation score, which is measured by matching model output against a "relational keywords dictionary" automatically constructed (using an LLM) from the model's own correct explanations . Experiments show this RL optimization provides significant gains, such as a +23.46% absolute F1 improvement on One-shot NYT29 and a 54% relative increase in human-rated explanation quality.

**Strengths:**

1. The paper provides strong empirical support through rich experimentation. Beyond baseline F1 comparisons, the authors include detailed ablation studies to validate their framework's components, an analysis of training dynamics to show the reward's behavior.
2. The HIT@DICT reward is a novel approach to supervising explanation quality without costly human-annotated data. The method of automatically constructing a "credit dictionary" by using an LLM to extract keywords from the model's own correct outputs is a clever, data-efficient way to create a fine-grained reward signal for jointly optimizing accuracy and explainability.

**Weaknesses:**

1. The task definition is unconventional. While framed as Relation Extraction (RE), the task is more accurately described as "Relation Matching"—a binary classification on whether entity pairs in two sentences share a relation. This setup is considerably simpler than traditional multi-class Relation Classification.

2. The authors appear to overlook highly relevant related work [1][2] concerning Relation Extraction using RLVR. A discussion of these papers seems necessary to properly position this work.

3. The performance attributed to HIT@DICT is debatable. The authors use a much stronger model (GPT-4o, per the paper) to analyze the student model's reasoning and create the high-quality keyword dictionary. This introduces an external dependency on a powerful model, creating an unfair advantage that is not available to the baseline methods.

4. The authors should consider adding a simpler, crucial baseline: SFT using only the final binary classification label (Yes/No), without any CoT generation. This is a more direct and lightweight approach, and its performance would provide a clearer context for the benefits of the proposed RL method.

[1]Li, Ran, et al. "Beyond path selection: Better LLMs for Scientific Information Extraction with MimicSFT and Relevance and Rule-induced (R2) GRPO." [2]Dai, Runpeng, et al. "R1-re: Cross-domain relation extraction with rlvr."

**Questions:**

NA

---

> ### Author Response · Authors · 2025-11-20
>
> Dear Reviewer,
>
> Thank you for your comments. We truly appreciate your thoughtful suggestions and guidance, which help us improve our work. In response to your review, we have made the following revisions:
>
> 1. Clarified why our setting is a multi-class Relation Extraction task, and explained why our RE setting is harder and more realistic. (Addressing Weakness 1)
> 2. Discussed the differences between our work and two recent RL-based RE methods. (Addressing Weakness 2)
> 3. Clarified the non-teacher role of GPT-4o and conducted drop-in replacement experiments showing that the dictionary can be reliably rebuilt using small open-source models. (Addressing Weakness 3)
> 4. Added SFT experiments and explained why SFT is not included as a baseline due to its low explainability. (Addressing Weakness 4)
>
> More details are provided below.

---

> ### Author Response · Authors · 2025-11-20
> **Response to Weakness 1 about our task definition**
>
> The Relation Extraction task in this paper is **(1) a multi-class RE task** framed under a realistic Few-shot RE setting [1]. The model must infer a relation by comparing a test sentence against several support sentences, without predefined labels or handcrafted label descriptions. This setting is **(2) harder** and more realistic than traditional multi-class RE.
>
> (1) **The Relation Extraction task we used in this paper is multi-class Relation Classification.** Although the intermediate step is pairwise matching, the task remains multi-class and is evaluated by whether the predicted relation matches the gold label. This pairwise formulation is also standard in Few-shot RE (e.g., SUMASK [2], Semantic Rule Matcher [3]).
> > **Task Definition**. For each relation, one support sentence is provided as its example. Given a test sentence, the model compares it with each support example and outputs Yes/No. If all outputs are No, the prediction is no_relation; otherwise, the corresponding relation is selected. The final prediction is evaluated using F1 against the gold label.
>
> (2) **Why this setting is harder.**
>
> * **No label semantics**. Traditional multi-class RE provides explicit label semantics or descriptions. In our setting, the model must infer a relational concept from a single support example and compare it to the one abstracted from the test sentence. This introduces two challenges discussed in Section 5.3: (a) the model may miss key semantic cues; (b) abstraction from support and test sentences is interdependent.
>
> * **Local (not global) relation semantics.** Traditional RE exposes the full label set, which helps it distinguish fine-grained differences between relations. In contrast, **in our task the relation semantics are local**, extracted only from the provided one support sentence.
>
>   > For example, consider the relations *org:country_of_headquarters* and *org:city_of_headquarters.* In a traditional multi-class setup, both labels are present in the candidate set, enabling the model to directly differentiate them. However, in our setting the model sees only one support example at a time, e.g., a support sentence expressing *org:country_of_headquarters*. If the test sentence actually expresses *org:city_of_headquarters*, the model lacks label definitions or a global label set. Consequently, naive LLMs tend to abstract a coarse concept such as *place_of_headquarters* from both sentences.
>
> * Evaluation results. This support-based formulation is known to be more challenging [1]. As reported in Appendix A.12 (Table 13), both 3B–7B open-source models and large mainstream LLMs perform poorly under this setting, confirming that realistic Few-shot RE remains difficult for LLMs.The dataset we use is the dev partition of one-shot TACRED.
>
>   | Model                    | Method                       | Setting | Precision | Recall | F1     |
>   | ------------------------ | ---------------------------- | ------- | --------- | ------ | ------ |
>   | Before RL-Qwen2.5-14B   | CogRE-Reasoning          | ICL     | 35.29%    | 39.47% | 37.27% |
>   | Deepseek-v3              | Direct-Answer                | ICL     | 35.62%    | 37.14% | 36.36% |
>   | Deepseek-v3              | Random-Reasoning-then-Answer | ICL     | 38.46%    | 21.43% | 27.52% |
>   | GPT-4o                   | Random-Reasoning-then-Answer | ICL     | 20.00%    | 23.00% | 21.33% |
>   | GPT-3.5-turbo            | Random-Reasoning-then-Answer | ICL     | 16.23%    | 36.00% | 22.32% |
>   | Qwen2.5-7B-Instruct      | Direct-Answer                | ICL     | 22.58%    | 9.21%  | 13.08% |
>   | Qwen2.5-7B-Instruct      | Random-Reasoning-then-Answer | ICL     | 3.42%     | 43.42% | 6.35%  |
>   | Qwen2.5-3B-Instruct      | Direct-Answer                | ICL     | 100%      | 0.00%  | 0.00%  |
>   | Qwen2.5-3B-Instruct      | Random-Reasoning-then-Answer | ICL     | 10.53%    | 7.89%  | 9.02%  |
>   | Phi-4-mini-Instruct (4B) | Direct-Answer                | ICL     | 0.88%     | 28.95% | 1.70%  |
>   | Phi-4-mini-Instruct (4B) | Random-Reasoning-then-Answer | ICL     | 1.65%     | 52.63% | 3.21%  |
>   | Llama-3.2-8B-Instruct    | Direct-Answer                | ICL     | 1.61%     | 39.47% | 3.09%  |
>   | Llama-3.2-8B-Instruct    | Random-Reasoning-then-Answer | ICL     | 0.43%     | 17.11% | 0.85%  |
>   | Mistral-7B-2v            | Direct-Answer                | ICL     | 5.00%     | 17.11% | 7.74%  |
>   | Mistral-7B-2v            | Random-Reasoning-then-Answer | ICL     | 0.95%     | 26.32% | 1.83%  |
>
> [1] Fahmida Alam, Md Asiful Islam, Robert Vacareanu, and Mihai Surdeanu. Towards realistic few-shot relation extraction: A new meta dataset and evaluation, 2024.
> [2] Guozheng Li, Peng Wang, and Wenjun Ke. Revisiting large language models as zero-shot relation extractors, 2023.
> [3] Robert Vacareanu, Fahmida Alam, Md Asiful Islam, Haris Riaz, and Mihai Surdeanu. Best of both worlds: A pliable and generalizable neuro-symbolic approach for relation classification, 2024b.

---

> ### Author Response · Authors · 2025-11-20
> **Response to Weakness 2 about two relevant recent works**
>
> Thank you for pointing out these papers! While both works make valuable contributions, there are significant differences from our task and method, as detailed below. We will add this discussion to the Related Work section.
>
> Regarding **R1-re: Cross-domain Relation Extraction with RLVR**, we note that:
>
> * **Task**: It adopts a **different problem setting** from ours. R1-re performs relation extraction under a fully supervised framework based on predefined annotation guidelines (label descriptions). And, constructing such guidelines is costly and often infeasible in new domains.  In contrast, our paper follows a more realistic and more challenging one-shot support-based scenario that does not rely on label descriptions or any human-crafted guidelines.
> * **Reward**: The RLVR reward used in R1-re supervises only the final prediction accuracy, whereas our approach introduces a novel Acc+Hit@Dict reward that evaluates both accuracy and the quality of explanations.
> * **Evaluation**: R1-re does not evaluate explanation quality after RL training, while our work includes a dual evaluation protocol, including both automatic metrics and human scoring.
>
> Regarding **Beyond Path Selection**, we note that:
>
> * **Task**: Their paper adopts an end-to-end two-stage information extraction task, including both named entity recognition and relation extraction. Our task is a realistic and challenging few-shot relation extraction setting, where the model should classify relations solely based on a small set of support sentences without any relation label semantics or relation label description.
> * **Different Capabilities of LLMs Under Study**: Their paper investigates how to enhance LLMs’ ability to generate composite schemas and factual information. Our work investigates how to enhance LLMs’ ability to abstract and distinguish relation labels in complex sentences without any descriptions of the labels.
> * **Training Paradigm**: Their method relies on supervised fine-tuning (SFT) as initialization followed by reinforcement learning. In construction, our approach directly performs reinforcement learning.
> * **Supervision labels for RL**: Their RL supervision depends on gold labels and human-crafted rules. Our reward leverages the model’s own high-quality outputs as supervision signals.
> * **RL training goals**: Their RL training aims to reward correct structure and grounding evidence. Ours focuses on aligning the models’ prediction and explanation with relation semantics.
>
> Therefore, both works are related but not directly comparable to our work. We will clarify this distinction in the final version.

---

> ### Author Response · Authors · 2025-11-20
> **Response to Weakness 3 about the Use of GPT-4o for Dictionary Construction**
>
> **(1) Building this dictionary is straightforward**: it is small (each relation includes only about 10 keywords) and can be created either manually from a few true-positive explanations or automatically using small or large LLMs.
>
> **(2) Drop-in replacement experiment**: To address this concern, we performed a drop-in replacement analysis by using Qwen2.5-14B instead of GPT-4o to extract keywords from explanations on the NYT29 dataset. We evaluate the effect from **two aspects:**
>
> * **Keyword Overlap. (91.95 %)**
>   We compare the overlap rate between keyword sets extracted by GPT-4o and Qwen2.5-14B. The two models produce highly consistent keyword sets, indicating that the extraction process is not sensitive to the choice of LLM.
>
>   To measure the consistency between the keyword sets generated by GPT-4o and Qwen2.5-14B, we use the **Overlap Coefficient**. Given two keyword sets $K_{\text{GPT4o}}$ and $K_{\text{Qwen}}$, the overlap is defined as:
>   $$
>   \text{Overlap} =
>   \frac{|K_{\text{GPT4o}} \cap K_{\text{Qwen}}|}
>   {\min(|K_{\text{GPT4o}}|, |K_{\text{Qwen}}|)}.
>   $$
>   For example, in the case shown in the following  table, we have:
>   $$
>   |K_{\text{GPT4o}} \cap K_{\text{Qwen}}| = 5,
>   \qquad
>   \min(|K_{\text{GPT4o}}|, |K_{\text{Qwen}}|) = 6,
>   $$
>   Across all 15 relations in the NYT29 training set, the two models produce highly consistent keyword sets. In total, we observe:
>   $$
>   |K_{\text{GPT4o}} \cap K_{\text{Qwen}}| = 80,
>   \qquad
>   \min(|K_{\text{GPT4o}}|, |K_{\text{Qwen}}|) = 87,
>   $$
>
>   $$
>   \text{Overlap} = \frac{80}{87} = 0.91954.
>   $$
>
>   **Table: The following table show the keywords for the label of  */location/location/contains* seperately by Qwen2.5-14B and GPT4o.** (We will give all 15 relations' keywords extracted seperately by Qwen2.5-14B and  GPT4o in Appendix in the final version of paper.)
>
>   | Qwen2.5-14B                     | GPT4o                                                        |
>   | ------------------------------- | ------------------------------------------------------------ |
>   | **/location/location/contains** | **/location/location/contains**                              |
>   | "city"                          | "city", # Case 1: Houston – Texas; Turin – Italy; Case 3: North Olmsted – Ohio; Shandaken – Ulster County; Case 4: Boise – Idaho |
>   | "location"                      | "location", # Case 2: New Rochelle – New York; Ballybunion – Ireland |
>   | "province"                      | "provincial", # Case 4: Changchun – China                    |
>   | "capital"                       | "capital", # Case 4: Changchun – China                       |
>   | "located"                       | "located", # Case 5: Rotterdam – Netherlands; Mankato – Minnesota |
>   |                                 | "contains"                                                   |
>   | "part"                          |                                                              |
>
>
>
> * **Performance of Our Method with a New Dictionary**.
>
>   We rebuild the entire dictionary using the Qwen2.5-14B extracted keywords and re-run the RL + Acc + Hit@Dict training. The resulting F1 score remains very close to the GPT-4o–based setup, showing that our method is robust under drop-in replacement.
>
>   | Dictionary Source                 | P     | R     | F1    |
>   | --------------------------------- | ----- | ----- | ----- |
>   | With GPT-4o Build Dictionary      | 63.34 | 38.78 | 48.11 |
>   | With Qwen2.5-14B Build Dictionary | 45.35 | 51.28 | 48.13 |
>
>
>
> **(3) The stronger model (GPT-4o) is used only as an extracting tool.** It does not perform any form of “analyzing the student model’s reasoning.” The good explanations are identified purely through string matching and rule-based filtering on True Positive predictions. During RL training, the keyword-based reward is also entirely rule-based, relying only on string matching, not on GPT-4o’s judgments.
>
> **(4) The concept of teacher–student:** There is no teacher–student setup. Common notions of “teacher models,” such as
>
> * teacher-generated rationales for distillation,
> * teacher-generated labels for supervision,
> * teacher scoring or evaluating student outputs, or
> * teacher guiding the student’s reasoning trajectory,
>
> do not apply here. GPT-4o does not supervise, evaluate, or guide the RL model at any stage. It only serves as a lightweight keyword extractor.

---

> ### Author Response · Authors · 2025-11-20
> **Response to Weakness 4 about a SFT baseline**
>
> We conducted two SFT experiments using the same training data as our RL settings: RL with Acc only and RL with Acc + Hit@Dict.
>
> - **SFT-direct-answer**: use only the final binary classification label (Yes/No), without any CoT generation.
> - **SFT-random-reasoning**: randomly generate reasoning followed by the final Yes/No label. We use reasoning generated by DeepSeek-v3 from correct predictions as the reasoning component in the training data.
>
> The results of both SFT experiments are as follows:
>
> | Method                   | Model | P      | R      | F1     |
> | ------------------------ | ----- | ------ | ------ | ------ |
> | SFT-random-reasoning (2) | Phi-4 | 8.22%  | 52.86% | 14.23% |
> | SFT-direct-answer (1)    | Phi-4 | 33.33% | 48.43% | 39.49% |
> | RL-Acc-only              | Phi-4 | 20.45% | 40.00% | 41.02% |
> | RL-Acc+Hit@Dict          | Phi-4 | 45.14% | 44.89% | 45.01% |
>
> Across both settings, the SFT models consistently underperform the RL models. In particular, the SFT-direct-answer variant (the one you mentioned) is strictly weaker than both RL settings for two reasons:
>
> * First, it provides **no explainability**: the model produces only a final Yes/No label, with no intermediate reasoning, making its learning objective far simpler than the reasoning-augmented framework discussed in our paper.
> * Second, its **F1 score** are **lower than those of any RL setting**, even though it is trained on exactly the same data.
>
> These two limitations, especially the absence of explanations, are the main reasons why we do not further emphasize this SFT setting in the paper.

---

> ### Author Response · Authors · 2025-11-26
>
> Dear Reviewer GtZe,
>
> We truly appreciate your engagement with our work.
>
> To ensure we make the best use of the remaining discussion time, we would like to kindly check whether our responses have addressed all of your concerns. If there are any remaining points that you would like us to clarify or further discuss, we would be more than happy to address them.
>
> Thank you again for your time and effort in reviewing our paper.
>
> Best regards,
> The Authors

---

### Official Review · Reviewer_kcwt · 2025-10-31

**Soundness:** 3
**Presentation:** 3
**Contribution:** 3
**Rating:** 4
**Confidence:** 3

**Summary:**

The paper proposes COGRE, a three-step, 'cognitively inspired; framework for one-shot relation extraction (RE): proposition chunking, keyword anchoring, and integrative reasoning. It adds an RL objective that sums an accuracy reward with HIT@DICT, a rule-based explanation reward built from a relational-keywords dictionary distilled by GPT-4o from true-positive LLM explanations (Alg. 1). On one-shot TACRED/NYT29, COGRE improves F1 over prompting baselines; adding RL with HIT@DICT further boosts F1 and yields shorter, human-preferred explanations.

**Strengths:**

S1) A clear lear, modular framework that operationalizes explainable RE with readable traces.

S2) I think this design is reasonably simple, and reproducible tht avoids costly LLM-as-a-judge at training time; formulas here are explicit.

S3) Solid empirical picture wuith consistent improvements across two LLMs and datasets.

**Weaknesses:**

W1) I believe the reward here may have a fatal flaw in that it maybe prioritizing label/keyword copying. If I understand the framework ((Alg 1)) correctly, RHIT@DICT explicitly uses ground truth labels for both sentences and includes decomposed label tokens in the dictionary, risking explanation gaming (keyword stuffing).

W2) Limited baselines -- Comparisons used by authors omit stronger LLM RE set-ups (e.g., richer in-context label descriptions or recent IE-aligned LLMs) beyond SUMASK and a neuro-symbolic classifier; scope could be broader.

W3) No variance over episode sampling. Authors sample 1,000 episodes per partition but report only point estimates (no CIs/std over different samplings/seeds), so stability is unclear.

W4) Hyperparameters fixed without sensitivity (L310-316). Reward weights w_entity=0.4, w_relation=1.0, N=5 and imbalanced-classification weights in Eq. (6) are “heuristically chosen,” but no ablation is provided. The authors should clarify how these values came about.

I think in totality this is good work, but evaluation could be made strengthened with stronger baselines, variance reporting, and reward-design robustness checks.

**Questions:**

Q1) Beyond length normalization in Eq. (5), how do you prevent models from inflating RHIT@DICT by repeating label tokens/synonyms? Any anti-duplication filters or hit caps?

Q2) Since you sample 1,000 episodes per split, can you report mean±std over ≥3 random episode samplings (and seeds) for Table 1?

Q3) How do results change if the dictionary is built with an open model (e.g., Phi) instead of GPT-4o? I think this analysis would be stronger if you included a drop-in replacement analysis.

Q4) For eq 8, what is θ_ref exactly (frozen backbone or SFT’d model)? Please report sensitivity to β.

---

> ### Author Response · Authors · 2025-11-20
>
> Dear Reviewer,
>
> Thank you for your recognition of this work. We truly appreciate your thoughtful suggestions and guidance, which help us improve our work. In response to your review, we have made the following revisions:
>
> 1. Clarified anti-gaming design choices, provided evidence of no keyword stuffing, and showed strong dependence between explanations and predictions. (Addressing Weakness 1 and Question 1)
> 2. Clarified the distinction between our realistic Few-Shot RE setting and the standard Few-Shot RE setup, and justified our baseline choices with detailed examples. (Addressing Weakness 2)
> 3. Reported variance across random episode samplings by providing mean ± std results and statistical significance tests (Addressing Weakness 3 & Question 2).
> 4. Explained hyperparameter choices and conduct sensitivity analysis experiments.(Addressing Weakness 4)
> 5. Conducted a drop-in replacement analysis using Qwen2.5-14B and demonstrated that our dictionary construction and model performance remain robust across different LLM choices. (Addressing Question 3)
> 6. Clarified the role of the reference model and explained our GRPO hyperparameter choices, including why β remains fixed. (Addressing Question 3)
>
> More details are provided below.

---

> > ### Author Response · Authors · 2025-11-20
> > **Response to Question 3 about Reference Model \(\theta_{\text{ref}}\) and GRPO Hyperparameters**
> >
> > In our implementation, \(\theta_{\text{ref}}\) represents the initial policy before GRPO optimization, which is the **frozen backbone (base model).** It remains unchanged throughout the RL stage and is used solely as the KL reference to prevent excessive policy drift. This follows the original GRPO formulation (Shao et al., 2024), where the reference model is defined as the initial policy.
> >
> > The coefficient β in GRPO balances the update magnitude against the KL regularization term, and we **keep it fixed as a penalty weight**. We **directly adopt the default configuration from the VeRL library [1] without further tuning**. During training, we **did not observe any instability, suggesting that the default value is sufficiently robust for our task.** We clarified this implementation detail in Section 4.
> >
> > [1] Sheng, Guangming, Chi Zhang, Zilingfeng Ye, Xibin Wu, Wang Zhang, Ru Zhang, Yanghua Peng, Haibin Lin, and Chuan Wu. **HybridFlow: A Flexible and Efficient RLHF Framework.** In *Proceedings of the Twentieth European Conference on Computer Systems (EuroSys ’25)*, pp. 1279–1297, 2025. ACM. DOI: 10.1145/3689031.3696075.
> > VeRL library

---

> ### Author Response · Authors · 2025-11-20
> **Response to Weakness 1 and Question 1 about explanation gaming**
>
> Thanks for your comments! We understand your concern. But our method does not risk label/keyword copying or explanation gaming (keyword stuffing). Our design prevents this in several ways:
>
> 1. **Prompt does not include relation labels and the relation keyword dictionary does not introduce label supervision**
>    Our method does not expose the model to gold relation labels: no relation label, label description, or label semantics appear in the prompt, and the model is not trained to output a label directly. The RL objective only rewards keywords that the model spontaneously generates by itself, rather than teaching it to memorize predefined label set. Thus, the dictionary may not cause label copying.
>
> 2. **Multiple reward design details are introduced to mitigate keyword stuffing:**
>
>    * **No repeated counting.**
>      As shown in Eq. (3), R_Hit@Dict iterates through unique keywords in the dictionary and adds a reward when the keyword appears. This means that every keyword contributes at most one unit of reward, regardless of how many times it is repeated in the output. Thus, repeating the same keyword cannot increase R_Hit@Dict, effectively preventing the model from hacking the reward by duplicating keywords.
>
>    * **Length penalty.**
>      As shown in Eq. (5), longer explanations are penalized, preventing reward hacking by appending many keywords.
>
>    * **Reward is applied only when the prediction is correct.**
>      Our released reward design code (anonymous GitHub) shows that we indeed implement the setting where R_Hit@Dict is added only when the prediction is correct. This prevents the model from stuffing keywords while still producing an incorrect prediction. We will further elaborate on this design in the final version.
>
> 3. **Empirical evidence shows no keyword-stuffing behavior.**
>    As shown in the case studies in Appendix A.4, the number of relation keywords generated in the "*Trained LLMs with HIT@DICT Output*" cases is entirely reasonable. Across all 160 *Trained LLMs with HIT@DICT Output* cases we evaluated, we did not observe keyword-stuffing behavior.
>
> 4. **Our evidence shows that there is strong functional dependence between the textual explanation and the final prediction.**
>
>    As shown in the ablation study in Section 5.4 and Table 3, removing any component of our frameword results in a clear performance drop, ranging from –1.66% to –21.51%. Moreover, different components affects the prediction in different ways: the keyword-anchoring step primarily improves precision, whereas the chunking and reasoning steps have a more noticeable impact on recall. These findings indicate that encouraging the model to output keywords through R_Hit@Dict helps LLMs to make correct predictions.

---

> ### Author Response · Authors · 2025-11-20
> **Response to Weakness 2 about baselines and our task setting**
>
> Our work focuses on the **realistic Few-Shot Relation Extraction (FSRE) task** [1], described in Sec. 1, Fig. 1, and Sec. 3.1. This task is **fundamentally different from** the **standard FSRE** setting used in recent LLM-based RE work, and is harder. Under this benchmark, the current SOTA is **Semantic Rule Matcher (2024) [2]**, included in Table 1. Our method achieves SOTA in this setting.
>
> **What “realistic” means.** In realistic FSRE, the model infers a relation by comparing a test sentence against several support sentences without predefined labels or label descriptions. This reflects real-world RE where many relations lack manual descriptions, and sentences are unlabeled.
>
> Below we explain why only Semantic Rule Matcher (neuro-symbolic classifier) and SUMASK (prompt-based)  are valid baselines.
>
> 1. Realistic FSRE removes label names and descriptions and is more challenging. Paper[1] note that “overall performance on this task is low.”  **Thus, comparing our method with standard FSRE task would be unfair.**
>    To illustrate this difficulty, Appendix A.12 (Table 13) evaluates many LLMs on the dev portion of one-shot TACRED.  All models perform poorly under this setting.
>
>    | Model                 | Method                       | Setting | Precision % | Recall % | F1 %  |
>    | --------------------- | ---------------------------- | ------- | ----------- | -------- | ----- |
>    | Before RL-Qwen2.5-14B | CogRE-Reasoning              | ICL     | 35.29       | 39.47    | 37.27 |
>    | Deepseek-v3           | Direct-Answer                | ICL     | 35.62       | 37.14    | 36.36 |
>    | Deepseek-v3           | Random-Reasoning-then-Answer | ICL     | 38.46       | 21.43    | 27.52 |
>    | GPT-4o                | Random-Reasoning-then-Answer | ICL     | 20.00       | 23       | 21.33 |
>    | GPT-3.5-turbo         | Random-Reasoning-then-Answer | ICL     | 16.2        | 36       | 22.32 |
>    | Qwen2.5-7B            | Direct-Answer                | ICL     | 22.58       | 9.21     | 13.08 |
>    | Qwen2.5-7B            | Random-Reasoning-then-Answer | ICL     | 3.42        | 43.42    | 6.35  |
>    | Qwen2.5-3B            | Direct-Answer                | ICL     | 100         | 0        | 0     |
>    | Qwen2.5-3B            | Random-Reasoning-then-Answer | ICL     | 10.53       | 7.89     | 9.02  |
>    | Phi-4-mini (4B)       | Direct-Answer                | ICL     | 0.88        | 28.95    | 1.70  |
>    | Phi-4-mini (4B)       | Random-Reasoning-then-Answer | ICL     | 1.65        | 52.63    | 3.21  |
>    | Llama-3.2-8B          | Direct-Answer                | ICL     | 1.61        | 39.47    | 3.09  |
>    | Llama-3.2-8B          | Random-Reasoning-then-Answer | ICL     | 0.43        | 17.11    | 0.85  |
>    | Mistral-7B-2v         | Direct-Answer                | ICL     | 5           | 17.11    | 7.74  |
>    | Mistral-7B-2v         | Random-Reasoning-then-Answer | ICL     | 0.95        | 26.32    | 1.83  |
>
> 2. Unfortunately, recent LLM-based RE methods **cannot be directly used in our realistic** FSRE setup, because (1) they are evaluated in different benchmark, and (2) require explicit relation labels, manual label descriptions, label-descriptive templates. All of these implementation requirements are not allowed in realistic FSRE. Thus using them would require extra information, making comparison invalid and unfair.
>    **Examples:**
>
>    * **CoT-ER[3]** requires a humancrafted CoT seed for each relation label, embedding label semantics; builds a large candidate pool and performs KNN retrieval.
>    * **QA4RE[4]** converts each label into a human-written question template that encodes label semantics. Moreover, QA4RE does not produce any reasoning and has therefore bad explainability.
>    * **Relation-Aware Prompting[5]** includes all relation labels, definitions, and descriptions in prompts.
>
> 3. **Baselines included in our paper.** We follow the evaluation method in Realistic FSRE[1], where the current SOTA is *Semantic Rule Matcher (2024.)* [2] (we include it in Table 1 and Section 4 Baselines). We also implement SUMASK under realistic FSRE constraints (no label descriptions) to show how prompting-based approaches behave in this setting.
>
> [1] Fahmida Alam, Md Asiful Islam, Robert Vacareanu, and Mihai Surdeanu. Towards realistic few-shot relation extraction: A new meta dataset and evaluation, 2024.
> [2] Robert Vacareanu, Fahmida Alam, Md Asiful Islam, Haris Riaz, and Mihai Surdeanu. Best of both worlds: A pliable and generalizable neuro-symbolic approach for relation classification, 2024b.
> [3] Xilai Ma, Jing Li, Min Zhang. Chain of Thought with Explicit Evidence Reasoning for Few-shot Relation Extraction, 2023.
> [4] Kai Zhang, Bernal Jiménez Gutiérrez, and Yu Su. Aligning Instruction Tasks Unlocks Large Language Models as Zero-Shot Relation Extractors, 2023.
> [5] Mahdi Rahimi, Razvan-Gabriel Dumitru, and Mihai Surdeanu. Relation-Aware Prompting Makes Large Language Models Effective Zero-shot Relation Extractors, 2025.

---

> ### Author Response · Authors · 2025-11-20
> **Response to Weakness 3 and Question 2 about Evaluation Variance Across Random Episode Samplings**
>
> Thank you for the comments. We now report results in the *mean ± std* format across three independently sampled test episode sets.
>
> Due to time limitations, we first re-evaluated **Phi-4** on the **NYT29** dataset and conducted **three independent random episode samplings**. For each sampled test set, we evaluated both **RL+Hit@Dict+Acc** and **RL+Acc** using the same protocol as Table 1. We report the corresponding **mean ± std** statistics and compute the **two-sided paired t-test p-values** between the two methods.
>
> The results show that performance is highly consistent across different random samplings, and the p-values confirm a statistically significant performance gap between the two methods.
>
> We will continue re-evaluating all settings reported in Table 1 and include the full *mean ± std* version in the camera-ready.
>
> **Raw Evaluation Results (Three Samplings)**
>
> (1) Phi-4 NYT29 — RL + Hit@Dict + Acc
>
> | Sample | P     | R     | F1    |
> | ------ | ----- | ----- | ----- |
> | 0      | 45.14 | 44.89 | 45.01 |
> | 1      | 46.21 | 45.05 | 45.62 |
> | 2      | 45.86 | 45.40 | 45.63 |
>
> (2) Phi-4 NYT29 — RL + Acc
>
> | Sample | P     | R     | F1    |
> | ------ | ----- | ----- | ----- |
> | 0      | 45.50 | 37.36 | 41.02 |
> | 1      | 45.67 | 37.50 | 41.19 |
> | 2      | 45.85 | 37.64 | 41.34 |
>
> **Aggregated Results (Mean ± Std) With Significance Test**
>
> | Model / Method              | P (mean ± std) | R (mean ± std) | F1 (mean ± std) | p-value (paired t-test) |
> | --------------------------- | -------------- | -------------- | --------------- | ----------------------- |
> | Phi-4 NYT29 RL+Hit@Dict+Acc | 45.74 ± 0.44   | 45.11 ± 0.21   | 45.42 ± 0.27    | 0.00047                 |
> | Phi-4 NYT29 RL+Acc          | 45.67 ± 0.18   | 37.50 ± 0.14   | 41.18 ± 0.16    | —                       |
>
> **How p-value Was Computed**
>
> We performed a two-sided paired t-test over the three sampled F1 scores:
>
> - RL+Hit@Dict+Acc F1 = [45.01, 45.62, 45.63]
> - RL+Acc F1 = [41.02, 41.19, 41.34]
>
> The resulting p-value is: p = 0.00047.
>
> A p-value < 0.001 indicates that RL+Hit@Dict+Acc significantly outperforms RL+Acc across episode samplings.

---

> ### Author Response · Authors · 2025-11-20
> **Response to Weakness 4 about Hyperparameter Choices and Sensitivity Analysis**
>
> Thank you for pointing this out! We explain how these values were chosen and present the corresponding hyperparameter-sensitivity experiments as follows:
>
> **(1) Source of the hyperparameter values.**
> These hyperparameters were initially set using simple heuristics.
> We set $W_{entity}$ = 0.4 and $W_{relation}$ = 1.0 based on the intuition that relation-level cues generally provide stronger discriminative power than entity-level signals.
> The choice of $N = 5$ was made after examining how the Hit@Dict reward behaves under different \(N\) values and selecting a balanced configuration that provides stable supervision without making the reward overly sparse.
> As discussed in Section 3.2 (Accuracy Reward), the class-imbalance weights in Eq. (6) follow Lin et al. (2019), *Deep Reinforcement Learning for Imbalanced Classification*. In our one-shot setting, each test instance is compared against \(K\) supports, where at most one is positive and the remaining ones are negative, naturally forming a 1:\(K\) imbalance. Following their formulation, we weight the reward such that correct *Yes* predictions receive higher scores and incorrect *Yes* predictions receive stronger penalties, effectively counteracting this imbalance.
>
> **(2) Hyperparameter-sensitivity experiments.**
> We additionally conduct sensitivity studies over these hyperparameters to verify that our method remains stable across a range of reasonable values.
>
> Due to time constraints, we limited this experiment to evaluating three values for each hyperparameter, including $W_{entity}$, $W_{relation}$, and $N$. If the paper is accepted, we will expand this analysis in the camera-ready version.
>
> To ensure controlled variation, all experiments were conducted using **Qwen2.5-14B-Instruct** on our sampled **NYT29** dataset (the same dataset used for all NYT29 training). We maintained the same RL training setup and sampling configuration as in the main experiments. In total, we tested **seven configurations**, and the results are reported in the following table.
>
> Across these seven hyperparameter configurations, the model’s performance remains highly stable. The F1 performance remains consistent at **48.12 ± 0.81**, with a **mean F1 of 48.12** and a **standard deviation of only 0.81**. This small variance indicates that our method is **robust to changes in weight settings** ( $W_{entity}$, $W_{relation}$, ) and the number of samples  $N$, and **does not rely on sensitive hyperparameter tuning**. Therefore, the observed improvements are attributable to the proposed method itself rather than fragile configuration choices.
>
> | Setting_No | W_entity | W_relation | N    | P     | R     | F1    |
> | ---------- | -------- | ---------- | ---- | ----- | ----- | ----- |
> | 1          | 0.4      | 1          | 5    | 63.34 | 38.78 | 48.11 |
> | 2          | 0.4      | 1          | 3    | 43.75 | 56.68 | 49.38 |
> | 3          | 0.4      | 1          | 7    | 44.06 | 52.13 | 47.76 |
> | 4          | 0.2      | 1          | 5    | 45.73 | 51.70 | 48.53 |
> | 5          | 0.6      | 1          | 5    | 45.61 | 48.72 | 47.12 |
> | 6          | 0.4      | 1.2        | 5    | 35.05 | 72.58 | 47.27 |
> | 7          | 0.4      | 0.8        | 5    | 47.34 | 48.01 | 47.67 |

---

> ### Author Response · Authors · 2025-11-20
> **Response to Question 3 about Rebuilding the Dictionary with Open-Source Models**
>
> **Building this dictionary is straightforward:** it is small (each relation includes only about 10 keywords) and can be created either manually from a few true-positive explanations or automatically using small or large LLMs.
>
> To address this concern, we performed **a drop-in replacement analysis** by using Qwen2.5-14B instead of GPT-4o to extract keywords from explanations on the NYT29 dataset. We evaluate the effect from two aspects:
>
> **(1) Keyword Overlap.**
> We compare the overlap rate between keyword sets extracted by GPT-4o and Qwen2.5-14B. **The two models produce highly consistent keyword sets, indicating that the extraction process is not sensitive to the choice of LLM.**
>
> To measure the consistency between the keyword sets generated by GPT-4o and Qwen2.5-14B, we use the **Overlap Coefficient**. Given two keyword sets $K_{\text{GPT4o}}$ and $K_{\text{Qwen}}$, the overlap is defined as:
>
> $$
> \text{Overlap} =
> \frac{|K_{\text{GPT4o}} \cap K_{\text{Qwen}}|}
> {\min(|K_{\text{GPT4o}}|, |K_{\text{Qwen}}|)}.
> $$
>
> **Table: The following table show the keywords for the label of  */location/location/contains* seperately by Qwen2.5-14B and GPT4o.** (We will give all 15 relations' keywords extracted seperately by Qwen2.5-14B and  GPT4o in Appendix in the final version of paper.)
>
> | Qwen2.5-14B                     | GPT4o                                                        |
> | ------------------------------- | ------------------------------------------------------------ |
> | **/location/location/contains** | **/location/location/contains**                              |
> | "city"                          | "city", # Case 1: Houston – Texas; Turin – Italy; Case 3: North Olmsted – Ohio; Shandaken – Ulster County; Case 4: Boise – Idaho |
> | "location"                      | "location", # Case 2: New Rochelle – New York; Ballybunion – Ireland |
> | "province"                      | "provincial", # Case 4: Changchun – China                    |
> | "capital"                       | "capital", # Case 4: Changchun – China                       |
> | "located"                       | "located", # Case 5: Rotterdam – Netherlands; Mankato – Minnesota |
> |                                 | "contains"                                                   |
> | "part"                          |                                                              |
>
> For example, in the case shown in the table, we have:
>
> $$
> |K_{\text{GPT4o}} \cap K_{\text{Qwen}}| = 5,
> \qquad
> \min(|K_{\text{GPT4o}}|, |K_{\text{Qwen}}|) = 6,
> $$
>
> and the complete example is provided in the appendix.
>
> **Across all 15 relations in the NYT29 training set,** the two models produce highly consistent keyword sets. In total, we observe:
> $$
> |K_{\text{GPT4o}} \cap K_{\text{Qwen}}| = 80,
> \qquad
> \min(|K_{\text{GPT4o}}|, |K_{\text{Qwen}}|) = 87,
> $$
>
> which yields an overall overlap of:
>
> $$
> \text{Overlap} = \frac{80}{87} = 0.91954.
> $$
> We will give all 15 relations' keywords extracted seperately by Qwen2.5-14B and  GPT4o in Appendix in the final version of paper.
>
> **(2) Performance of Our Method with a New Dictionary.**
>
> We rebuild the entire dictionary using the Qwen2.5-14B extracted keywords and re-run the RL + Acc + Hit@Dict training. The resulting F1 score remains very close to the GPT-4o–based setup, showing that our method is robust under drop-in replacement.
>
> | Dictionary Source                 | P     | R     | F1    |
> | --------------------------------- | ----- | ----- | ----- |
> | With GPT-4o Build Dictionary      | 63.34 | 38.78 | 48.11 |
> | With Qwen2.5-14B Build Dictionary | 45.35 | 51.28 | 48.13 |

---

> ### Author Response · Authors · 2025-11-26
>
> Dear Reviewer kcwt,
>
> Thank you once again for your detailed comments on our submission.
>
> To ensure we make the best use of the remaining discussion time, we would like to kindly check whether our responses have addressed all of your concerns. If there are any remaining points that you would like us to clarify or further discuss, we would be more than happy to address them.
>
> Thank you again for your time and effort in reviewing our paper.
>
> Best regards,
> The Authors

---

### Official Review · Reviewer_L1ac · 2025-11-01

**Soundness:** 3
**Presentation:** 2
**Contribution:** 4
**Rating:** 6
**Confidence:** 4

**Summary:**

This paper proposes a framework called COGRE, aiming to improve task accuracy and explainability for one-shot RE. Its core components include a three-step reasoning mechanism mimicking cognitive psychology (semantic chunking, keyword anchoring, and integrative reasoning) and an optimization process based on RL. The latter employs an innovative HIT@DICT reward function, which combines accuracy rewards with an explanation quality reward based on an automatically constructed relation keyword dictionary using LLM. As the result, COGRE is validated on the one-shot TACRED and NYT29 datasets, using Qwen2.5-15B-Instruct and Phi-4. It improves accuracy with balanced precision and recall and with the RL+HIT@DICT reward, the F1 score improves by an absolute 23.46%. Human evaluation shows that the relation keywords generated by the best model are highly aligned with the golden labels, resulting in a relative 54% improvement in explanation quality score.

**Strengths:**

1. The paper innovatively frames one-shot RE as a reasoning and explanation task, by designing a reasoning mechanism integrating cognitive psychology and the HIT@DICT reward function with a LLM-constructed dictionary, which simultaneously boost accuracy and explanation quality and can be transferred to other LLM reasoning tasks.
2. COGRE addresses two major issues in the RE field, including the lack of data in low-resource scenarios and supervision in language-based explanations. COGRE does not rely on the number of labeled samples, requiring only one supporting sentence/relation. Through a three-step reasoning mechanism, it generates traceable explanations, solving the trust problem of the black-box problem.
3. The paper details the experimental setup, clearly explaining the basic model, hyperparameters, hardware environment, and training time. It also provides training and testing data, as well as prompt templates, which will help researchers in related fields to further expand upon this.

**Weaknesses:**

1. The paper contains inconsistencies in describing prior work and lacks clarity in theoretical formalizations. In the Abstract, the paper frame its contribution as addressing the "lack of supervision for language-based explanations in traditional RE" that acknowledges prior work may have attempted language-based explanations but lacked supervision for them. However, in the Related Work, it overstate this by claiming "these approaches have limited explainability due to the lack of language-based explanations", which contradicts the actual state of the field.
2. The experimental setup undermines the paper’s claims of state-of-the-art (SOTA) performance and real-world applicability. The paper only compares against a 2023 prompt-based baseline(e.g., SUMASK (Li et al., 2023)). However, RE with LLMs is a rapidly evolving field these two years. Without comparing to recent works, the paper’s claim of being "SOTA" is unsubstantiated.
3. The paper’s evaluation of language-based explainability is limited to surface-level metrics, failing to validate the fidelity of explanations (whether explanations truly reflect the model’s reasoning).

**Questions:**

1. In Section 3.1, the paper defines s_1 and s_2 as "two input sentences", which is scattered across the text (not explicitly tied to Equation (1)) and lacks explicit labeling. For example, " s_1 denotes the support sentence and s_2 denotes the test sentence"), leading to confusion for readers.
2. The theoretical introduction of the three-step reasoning (chunking → anchoring → integration) is overly brief. For example, it should explain how chunking aligns with Miller’s 1956 theory beyond a passing citation or how each step mitigates LLM hallucinations with mechanistic details.
3. The paper’s core motivation is to address explainability in "high-stakes domains (healthcare, law, finance)" (Section 1). However, it only evaluates on general-domain datasets (one-shot TACRED and NYT29). This misalignment raises critical questions about real-world applicability. For example, can COGRE accurately chunk domain-specific terms in healthcare, or generate meaningful explanations for professional relations? Without testing on domain-specific datasets, the paper fails to validate its intended use case.
4. The paper uses fixed hyperparameters for the HIT@DICT reward and GRPO optimization but provides no sensitivity analysis. For instance, how do changes in ω_entity and ω_relation affect F1 and explanation quality?
5. The core problem the paper aims to solve is "the lack of supervision for language-based explanations", which implicitly requires ensuring that explanations are causally linked to the model’s decisions (not just textually aligned with gold labels). However, the paper only evaluates explanations via human subjective scores (3-point Likert scale) and keyword matching rates with the dictionary. If keywords are removed from the matching dictionary, can the model still correctly identify the relations?

---

> ### Author Response · Authors · 2025-11-20
>
> Dear Reviewer,
>
> Thank you for your recognition of this work. We truly appreciate your thoughtful suggestions and guidance, which help us improve our work. In response to your review, we have made the following revisions:
>
> 1. Clarified the context of “lack of supervision for language-based explanations” and
>    “lack of language-based explanations.” (Addressing Weakness 1)
> 2. Clarified the difference between our task setting and the standard Few-Shot RE setup, and explained our baseline selection. (Addressing Weakness 2)
>
> 3. Expanded the evaluation of explanations and clarified the fidelity between explanations and predictions. (Addressing Weakness 3 and Question 5)
>
> 4. Added explicit definitions of \(s_1\) and \(s_2\), expanded the introduction of the
>    three-step reasoning process, and clarified our motivation regarding high-stakes domains.  (Addressing Question 1, Question 2, and Question 3)
>
> 5. Conducted hyperparameter sensitivity experiments. (Addressing Question 4)
>
> More details are provided below.

---

> > ### Author Response · Authors · 2025-11-20
> > **Response to Question 2 about the introduction of the three-step reasoning and the mechanistic details of each step**
> >
> > Due to space limits, we will expand the description of how each step inspired by each cognitive theory in the final version.
> >
> > Please note that our method provides language- and cognitive-based explanations rather than mechanistic interpretability. We discuss the impact of each explainability component on mitigating hallucinations in Section 3.1, and analyze error patterns without our CogRE and after with our CogRE in Section 5.3.
> >
> > > Section 3.1: Proposition Chunking ensures that the LLMs’ analysis process starts with compressed propositions instead of long sequences of tokens. And Keywords Anchoring grounds the LLMs’ relation-matching reasoning in the original sentence and the extracted propositions. Integrative Reasoning giving a coherent logical chain.
> > >
> > > Section 5.3: First, failing to focus on semantics that truly convey relation…. Second, failing to align with the abstraction level defined in the RE human-annotation schema. And our CogRE mitigates these two failure patterns.

---

> ### Author Response · Authors · 2025-11-20
> **Response to Weakness 1 about two "lack of..." phrases**
>
> Thanks for this comment. The two “lack of…” phrases refer to two different concepts in different contexts.
>
> **In the Related Work section (2.1)**, the sentence “However, due to the lack of language-based explanations, these approaches have limited explainability” refers to the feature-based methods, neural networks, and small pre-trained language models discussed earlier. These methods rarely produce any language-based explanations. In addition, post-hoc approaches such as LIME [1] and SHAP [2], which generate feature-level explanations after the model’s prediction, perform poorly on RE because their explanations are disconnected from the actual RE model. This can be seen in Table 4 and Table 5 of *It Takes Two Flints to Make a Fire: Multitask Learning of Neural Relation and Explanation Classifiers* [3]. At this point in our paper, we have not yet introduced LLM-based reasoning; that content appears in the following subsection (Section 2.2).
>
> **In contrast, the Abstract present our work from a broader perspective**. The sentence “Our framework addresses the lack of supervision for language-based explanations in traditional RE by promoting outputs that include important relation keywords” is tended to emphasize that our method CogRE not only augments models with language-based explanations, but also makes explanations more structured, verifiable, and directly supervisable through the use of relation keywords.
>
> Thus,  **the two sentences describe two distinct levels of explainability**, and they are not contradictory. **We are happy to further clarify this in the final paper.**
>
> [1] Marco Tulio Ribeiro, Sameer Singh, Carlos Guestrin. “Why Should I Trust You? : Explaining the Predictions of Any Classifier
>
> [2] Scott M. Lundberg & Su-In Lee. “A Unified Approach to Interpreting Model Predictions”. 2017.
>
> [3] Zheng Tang, Mihai Surdeanu. It Takes Two Flints to Make a Fire: Multitask Learning of Neural Relation and Explanation Classifiers.

---

> ### Author Response · Authors · 2025-11-20
> **Response to Weakness 2 about comparison with recent RE baselines**
>
> Our work focuses on the **realistic Few-Shot Relation Extraction (FSRE) task** [1], described in Sec. 1, Fig. 1, and Sec. 3.1. This task is **fundamentally different from** the **standard FSRE** setting used in recent LLM-based RE work, and is harder. Under this benchmark, the current SOTA is **Semantic Rule Matcher (2024) [2]**, included in Table 1. Our method achieves SOTA in this setting.
>
> **What “realistic” means.** In realistic FSRE, the model infers a relation by comparing a test sentence against several support sentences without predefined labels or handcrafted label descriptions. This reflects real-world RE scenarios where many relations lack manual descriptions, and many sentences are unlabeled.
>
> Below we explain **why only Semantic Rule Matcher and SUMASK are valid baselines**.
>
> 1. Realistic FSRE removes label names and descriptions and is **more challenging**. Paper[1] note that *“overall performance on this task is low.”*  **Thus, comparing our method with standard FSRE task would be unfair.**
>    To illustrate this difficulty, Appendix A.12 (Table 13) evaluates many LLMs on the dev portion of one-shot TACRED.  All models perform poorly under this setting.
>
>    | Model                 | Method                       | Setting | Precision % | Recall % | F1 %  |
>    | --------------------- | ---------------------------- | ------- | ----------- | -------- | ----- |
>    | Before RL-Qwen2.5-14B | CogRE-Reasoning              | ICL     | 35.29       | 39.47    | 37.27 |
>    | Deepseek-v3           | Direct-Answer                | ICL     | 35.62       | 37.14    | 36.36 |
>    | Deepseek-v3           | Random-Reasoning-then-Answer | ICL     | 38.46       | 21.43    | 27.52 |
>    | GPT-4o                | Random-Reasoning-then-Answer | ICL     | 20.00       | 23       | 21.33 |
>    | GPT-3.5-turbo         | Random-Reasoning-then-Answer | ICL     | 16.2        | 36       | 22.32 |
>    | Qwen2.5-7B            | Direct-Answer                | ICL     | 22.58       | 9.21     | 13.08 |
>    | Qwen2.5-7B            | Random-Reasoning-then-Answer | ICL     | 3.42        | 43.42    | 6.35  |
>    | Qwen2.5-3B            | Direct-Answer                | ICL     | 100         | 0        | 0     |
>    | Qwen2.5-3B            | Random-Reasoning-then-Answer | ICL     | 10.53       | 7.89     | 9.02  |
>    | Phi-4-mini (4B)       | Direct-Answer                | ICL     | 0.88        | 28.95    | 1.70  |
>    | Phi-4-mini (4B)       | Random-Reasoning-then-Answer | ICL     | 1.65        | 52.63    | 3.21  |
>    | Llama-3.2-8B          | Direct-Answer                | ICL     | 1.61        | 39.47    | 3.09  |
>    | Llama-3.2-8B          | Random-Reasoning-then-Answer | ICL     | 0.43        | 17.11    | 0.85  |
>    | Mistral-7B-2v         | Direct-Answer                | ICL     | 5           | 17.11    | 7.74  |
>    | Mistral-7B-2v         | Random-Reasoning-then-Answer | ICL     | 0.95        | 26.32    | 1.83  |
>
> 2. Unfortunately, recent LLM-based RE methods **cannot be directly used in our realistic** **FSRE** setup, because (1) they are evaluated in different benchmark, and (2) require explicit relation labels, human-written label descriptions, label-descriptive templates. All of these implementation requirements are forbidden in realistic FSRE. Thus using them would require extra information, making comparison unfair.
>    **Examples:**
>
>    * **CoT-ER[3]** requires a manually written CoT seed for *each* relation label, embedding label semantics; builds a large candidate pool and performs KNN retrieval.
>    * **QA4RE[4]** converts each label into a human-written question template that encodes label semantics. Moreover, QA4RE does not produce any reasoning and has therefore bad explainability.
>    * **Relation-Aware Prompting[5]** includes all relation labels, definitions, and descriptions in prompts.
>
> 3. **Baselines included in our paper.** We follow the evaluation method in Realistic FSRE[1], where the current SOTA is *Semantic Rule Matcher (2024.)* [2] (we include it in Table 1 and Section 4 Baselines). We also implement **SUMASK** under realistic FSRE constraints (no label descriptions) to show how prompting-based approaches behave in this setting.
>
> [1] Fahmida Alam, Md Asiful Islam, Robert Vacareanu, and Mihai Surdeanu. Towards realistic few-shot relation extraction: A new meta dataset and evaluation, 2024.
> [2] Robert Vacareanu, Fahmida Alam, Md Asiful Islam, Haris Riaz, and Mihai Surdeanu. Best of both worlds: A pliable and generalizable neuro-symbolic approach for relation classification, 2024b.
> [3] Xilai Ma, Jing Li, Min Zhang. Chain of Thought with Explicit Evidence Reasoning for Few-shot Relation Extraction, 2023.
> [4] Kai Zhang, Bernal Jiménez Gutiérrez, and Yu Su. Aligning Instruction Tasks Unlocks Large Language Models as Zero-Shot Relation Extractors, 2023.
> [5] Mahdi Rahimi, Razvan-Gabriel Dumitru, and Mihai Surdeanu. Relation-Aware Prompting Makes Large Language Models Effective Zero-shot Relation Extractors, 2025.

---

> ### Author Response · Authors · 2025-11-20
> **Response to Weakness 3 and Question 5 about our evaluation of explanations and the fidelity of explanations**
>
> We would also like to clarify that our evaluation goes beyond purely surface-level metrics. We do have **some evidence** to support that **there is strong functional and causal dependence between the textual explanation and prediction**.
>
> 1. **The textual explanation quality directly affects the accuracy of the RE system**. As shown in Section 5.4 Ablation study and Table 3, removing any step from our framework leads to a clear performance drop, ranging from –1.66% to –21.51%. Furthermore, removing different steps affects the prediction results from different aspects. The keywords anchoring step primarily contributes to precision, while the chunking and reasoning steps affect more apparent recall.
>
> 2. **We find that textual explanations quality and prediction accuracy improve in parallel.**
>
>    *  After training, **both metrics increase** together: accuracy improves by +23.46% (absolute) and the human-evaluation score increases by +54% (relative).
>
>    * The **case studies** in Appendix A.4 further **illustrate this stepwise dependence**. Each explanation step directly affects the next, and an early error propagates to prediction. For example, in **Case 1**: (The same phenomenon is observed in the other cases as well.)
>
>      In *Vanilla LLMs Output*, the model produces a low-quality Relation_Summarization_1, which directly causes the keyword-anchoring step to fail and ultimately leads to an incorrect prediction.
>      In *Trained LLMs with Acc. Output*, incorrect relation keywords again lead to an incorrect prediction.
>      In contrast, in *Trained LLMs with HIT@DICT Output*, both relation summarizations and the relation keyword are correct, and the model produces the correct final prediction.
>
>    * We also **performed additional experiments** to **detect the inconsistency between explanations and predictions**. We used simple string-matching to detect patterns that explanations explicitly state that the two sentences express *different* relations (“different/differ/difference”) while the model predicts *Yes*; or explanations state they are *similar* while the model predicts *No*. We did not observe such inconsistencies in the final trained model. Our released reward-design code (anonymous GitHub) also includes an extract_differ function.
>
> **For the further concept of interpretability,** one could introduce Sparse Autoencoders to study causal mechanisms. However, **the goal of this paper is to improve both the accuracy and the explainability (the quality of textual explanations)**, aiming for a more accurate and **explainable** RE system. We do have ablation experiments to show **strong functional and causal dependence between the textual explanation and prediction, but we do not focus on investigating the mechanism of how LLMs make decisions**. Thank you again for raising this point! It is indeed an interesting and broader research question, and we will include it as a direction for future work.

---

> ### Author Response · Authors · 2025-11-20
> **Response to Question 1 about the definition of s_1 and s_2**
>
> Thank you for pointing this out! We will add a sentence in Section 3.1 to label them.
>
> > Let the input be denoted as \( x = (s_1, s_2) \), where \(s_1\) is the support sentence and \(s_2\) is the test sentence.

---

> ### Author Response · Authors · 2025-11-20
> **Response to Question 3 about the lack of high-stakes domains datasets**
>
> We’d like to clarify that the references to high-stakes domains(healthcare, law, finance) are used as general motivation for the importance of explainability. We list some published or top-tier papers, which similarly motivate explainability through high-stakes applications while evaluating on standard general-domain benchmarks: [1], [2], [3], [4], [5].
>
> Nevertheless, we share the reviewer’s concerns about over claiming. We will adjust the text in the paper to make it clear that in this initial work we focus on open-domain datasets.
>
>
>
> [1] Robert Vacareanu, Fahmida Alam, Md Asiful Islam, Haris Riaz, and Mihai Surdeanu. Best of both worlds: A pliable and generalizable neuro-symbolic approach for relation classification, 2024b.
>
> [2] Zheng Tang and Mihai Surdeanu. It takes two flints to make a fire: Multitask learning of neural relation and explanation classifiers. Computational Linguistics, 49:117–156, 2023. ISSN 1530-9312. doi: 10.1162/coli a 00463.
>
> [3] Xiusi Chen, Shanyong Wang, Cheng Qian, Hongru Wang, Peixuan Han, and Heng Ji. DecisionFlow: Advancing Large Language Model as Principled Decision Maker, 2025.
>
> [4] Arka Dutta, Sujan Dutta, Rijul Magu, Soumyajit Datta, Munmun De Choudhury, and Ashiqur R. KhudaBukhsh. What About the Scene with the Hitler Reference? HAUNT: A Framework to Probe LLMs' Self-consistency Via Adversarial Nudge, 2025.
>
> [5] Yaoyuan Zhang, Aishan Liu, Zonghao Ying, Xianglong Liu, Jiangfan Liu, Yisong Xiao, and Qihang Zhang. Uncovering Strategic Egoism Behaviors in Large Language Models, 2025.

---

> ### Author Response · Authors · 2025-11-20
> **Response to Question 4 about the  Hyperparameter Sensitivity Analysis**
>
> Due to time constraints, we limited this experiment to evaluating three values for each hyperparameter, including $W_{entity}$, $W_{relation}$, and $N$. If the paper is accepted, we will expand this analysis in the camera-ready version.
>
> To ensure controlled variation, all experiments were conducted using **Qwen2.5-14B-Instruct** on our sampled **NYT29** dataset (the same dataset used for all NYT29 training). We maintained the same RL training setup and sampling configuration as in the main experiments. In total, we tested **seven configurations**, and the results are reported in the following table.
>
> Across these seven hyperparameter configurations, the model’s performance remains highly stable. The F1 performance remains consistent at **48.12 ± 0.81**, with a mean F1 of 48.12 and a standard deviation of only 0.81. This small variance indicates that our method is **robust to changes in weight settings** ( $W_{entity}$, $W_{relation}$, ) and the number of samples  $N$, and **does not rely on sensitive hyperparameter tuning**. Therefore, the observed improvements are attributable to the proposed method itself rather than fragile configuration choices.
>
> | Setting_No | W_entity | W_relation | N         | P     | R     | F1    |
> | ---------- | -------- | ---------- | ---- | --------- | ----- | ----- |
> | 1          | 0.4      | 1          | 5    | 63.34 | 38.78 | 48.11 |
> | 2          | 0.4      | 1          | 3    | 43.75 | 56.68 | 49.38 |
> | 3          | 0.4      | 1          | 7    | 44.06 | 52.13 | 47.76 |
> | 4          | 0.2      | 1          | 5    | 45.73 | 51.70 | 48.53 |
> | 5          | 0.6      | 1          | 5    | 45.61 | 48.72 | 47.12 |
> | 6          | 0.4      | 1.2        | 5    | 35.05 | 72.58 | 47.27 |
> | 7          | 0.4      | 0.8        | 5    | 47.34 | 48.01 | 47.67 |

---

### Author Response · Authors · 2025-12-02
**General Response 2: Key Points in this Work (3)**

## Weakness

1. Reviewers L1ac and kcwt raise concerns about hyperparameter sensitivity and method stability
   We conducted comprehensive sensitivity analyses on key hyperparameters to evaluate the robustness of our method. All experiments were performed using Qwen2.5-14B-Instruct on the NYT29 dataset under the same RL training setup.

   | Setting No. | $W_{entity}$ | $W_{relation}$ | $N$  | Precision (%) | Recall (%) | F1 (%) |
   | ----------- | ------------ | -------------- | ---- | ------------- | ---------- | ------ |
   | 1           | 0.4          | 1.0            | 5    | 63.34         | 38.78      | 48.11  |
   | 2           | 0.4          | 1.0            | 3    | 43.75         | 56.68      | 49.38  |
   | 3           | 0.4          | 1.0            | 7    | 44.06         | 52.13      | 47.76  |
   | 4           | 0.2          | 1.0            | 5    | 45.73         | 51.70      | 48.53  |
   | 5           | 0.6          | 1.0            | 5    | 45.61         | 48.72      | 47.12  |
   | 6           | 0.4          | 1.2            | 5    | 35.05         | 72.58      | 47.27  |
   | 7           | 0.4          | 0.8            | 5    | 47.34         | 48.01      | 47.67  |

   The mean F1 is 48.12% with only 0.0081 of standard deviation. The results indicates that our method exhibits high stability across varying hyperparameter configurations, confirming that improvements are not dependent on fragile tuning.


2. Reviewer reviewer Pg7E request the comparison with state-of-the-art LLMs like GPT-4o and GPT-3.5.
   We fisrt clarify that the reason why we do not include the comparison with GPT-seris models is that they are closed-source with low reproducebility, high cost and low academic transparency.
   However, in the rebuttal, in response to reviews' requests, we further evaluated multiple mainstream LLMs under our realistic few-shot RE setting on the one-shot TACRED dev partition. Results are summarized below:

   | Model                 | Method                       | Setting | Precision (%) | Recall (%) | F1 (%) |
   | --------------------- | ---------------------------- | ------- | ------------- | ---------- | ------ |
   | Before RL-Qwen2.5-14B | CogRE-Reasoning              | ICL     | 35.29         | 39.47      | 37.27  |
   | DeepSeek-v3           | Direct-Answer                | ICL     | 35.62         | 37.14      | 36.36  |
   | DeepSeek-v3           | Random-Reasoning-then-Answer | ICL     | 38.46         | 21.43      | 27.52  |
   | GPT-4o                | Random-Reasoning-then-Answer | ICL     | 20.00         | 23.00      | 21.33  |
   | GPT-3.5-turbo         | Random-Reasoning-then-Answer | ICL     | 16.23         | 36.00      | 22.32  |
   | Qwen2.5-7B-Instruct   | Direct-Answer                | ICL     | 22.58         | 9.21       | 13.08  |
   | Qwen2.5-7B-Instruct   | Random-Reasoning-then-Answer | ICL     | 3.42          | 43.42      | 6.35   |
   | Qwen2.5-3B-Instruct   | Direct-Answer                | ICL     | 100.00        | 0.00       | 0.00   |
   | Phi-4-mini (4B)       | Direct-Answer                | ICL     | 0.88          | 28.95      | 1.70   |
   | Llama-3.2-8B-Instruct | Direct-Answer                | ICL     | 1.61          | 39.47      | 3.09   |
   | Mistral-7B-2v         | Direct-Answer                | ICL     | 5.00          | 17.11      | 7.74   |

   We find that even powerful models like GPT-4o and GPT-3.5-turbo perform poorly under our challenging realistic few-shot setting. The proposed method  substantially outperforms these models, demonstrating its substantial contribution in a setting where existing LLMs struggle.

3. Reviewer kcwt concerns the risk of reward hacking or keyword stuffing.
   We fully agree this concern and we do have already designed multiple mechanisms to prevent reward hacking in our work. Here are some key points:

   - we include several normlization in the reward function, such as

     > * No repeated counting: Each unique keyword contributes at most one unit of reward (Eq. 3).
     > * Length penalty: Longer explanations are penalized (Eq. 5).

   - The keywords-based reward is conditioned on correct prediction. In more details, R_Hit@Dict is added only if the prediction is correct.

   - We have conducted extensive validation, demonstrating that the proposed Dict@k reward function is effective in improving the reasoning ability rather than a keyword stuffing behavior.

     > Case studies (Appendix A.4) show no keyword-stuffing behavior across 160 evaluated cases.
     > Ablation study (Section 5.4, Table 3) confirms that removing any reasoning component leads to performance drops (-1.66% to -21.51%), indicating functional dependence between explanations and predictions.

---

### Author Response · Authors · 2025-12-02
**General Response 2: Key Points in this Work (2)**

## Misunderstandings

We summarize the misunderstandings we have solved as below.

1. Clarification on dictionary construction.

   > Reviewer Concern: Some reviewers requested further details on how the relational keyword dictionary is constructed.
   > We have clarified in Section 3.2 (lines 229–237) and in the Appendix that the dictionary is built by prompting an LLM to extract keywords from correctly predicted positive examples. The process is fully described with pseudocode and implementation steps, and reviewers L1ac and kcwt have acknowledged that the approach is clear and reproducible.

2. Model-Agnostic Nature of Dictionary Construction

   > Reviewer Concern: There was a misconception that the dictionary relies exclusively on GPT-4.
   > Our Response: The dictionary construction is model-agnostic. As demonstrated in response to Reviewer kcwt, we rebuilt the dictionary using the open-source model Qwen2.5-14B and observed high keyword overlap (over 91%) with the GPT-4o version. Performance remained stable, confirming that the method does not depend on a specific LLM.

3. Evaluation of Reasoning Ability Beyond End-to-End Metrics

   > Reviewer Concern: Reviewer L1ac questioned whether end-to-end metrics alone can capture improvements in reasoning ability.
   > Our Response: We agree that task accuracy alone is insufficient. In addition to Precision, Recall, and F1, we conducted fine-grained human evaluation to assess explanation quality and reasoning consistency. As reported in Section 5.3, models trained with our framework showed a 54% relative improvement in human-rated explanation quality, indicating a tangible enhancement in structured reasoning.

4. Generalizability of the HIT@DICT Reward

   > Reviewer Concern: Reviewer Pg7E raised concerns about whether the reward generalizes to unseen relations.
   > Our Response: As discussed in Section 4 and validated experimentally, our model is evaluated under an out-of-distribution setting where test relations are unseen during training. Results show that the model maintains strong performance, demonstrating that the reward encourages learning transferable relational patterns rather than memorizing fixed keyword sets.  Partticularly, the human evaluation (seen in Section 5.3 line 446-448, and 4th contribution in Introduction) shows that models trained with our reward exhibited better alignment with unseen label.

5. Comparison with recent RE baselines

   > Reviewer Concern: Reviewer L1ac and kcwt raised concerns about the baselines and our task setting.
   >
   > Our Response: Our work focuses on the **realistic Few-Shot Relation Extraction (FSRE) task** [1], described in Sec. 1, Fig. 1, and Sec. 3.1. This task is **fundamentally different from** the **standard FSRE** setting used in recent LLM-based RE work, and is harder. Under this benchmark, the current SOTA is Semantic Rule Matcher (2024) [2], included in Table 1. Our method achieves SOTA in this setting.
   >
   > **What “realistic” means.** In realistic FSRE, the model infers a relation by comparing a test sentence against several support sentences without predefined labels or handcrafted label descriptions. This reflects real-world RE scenarios where many relations lack manual descriptions, and many sentences are unlabeled.
   >
   > **Why only Semantic Rule Matcher and SUMASK are valid baselines**. (1) Realistic FSRE removes label names and descriptions and is more challenging. (2) recent LLM-based RE methods cannot be directly used in our realistic FSRE setup, because they are evaluated in different benchmark, and require explicit relation labels, human-written label descriptions, label-descriptive templates. (3) We also implement SUMASK under realistic FSRE constraints (no label descriptions) to show how prompting-based approaches behave in this setting.s

---

### Author Response · Authors · 2025-12-02
**General Response 2: Key Points in this Work (1)**

Below, we summarize the key points of this work, including the advantages mentioned by reviewers, the misunderstanding we clarified, and the disadvantages we have solved.

## Strengths

We summarize key strengths recognized by reviewers as below.

1. Innovative Framework for Explainable RE
   Our work reformulates one-shot relation extraction as a reasoning and explanation task, integrating principles from cognitive psychology into a structured reasoning mechanism. This approach not only enhances explainability but also aligns with human-like reasoning processes. [Reviewer L1ac]

2. Novel Reward Function for Unsupervised Explanation Supervision
   We introduce the HIT@DICT reward, a novel mechanism that enables supervision of explanation quality without relying on costly human-annotated data. By automatically constructing a relational keyword dictionary, our method provides fine-grained feedback on explanation quality, like relevence and conciseness. [Reviewers L1ac、GtZe]

3. Comprehensive and Rigorous Experimental Validation
   Our method is extensively evaluated across multiple datasets and model architectures, demonstrating consistent improvements in both accuracy and explainability. These results underscore the practical utility and robustness of our framework, contributing meaningfully to the field of explainable AI. [All reviewers]

---

### Author Response · Authors · 2025-12-02
**General Response 1: Summary of the Rebuttal Process**

Dear Area Chair, Senior Area Chair, and Program Chairs,

We sincerely thank all reviewers for their thoughtful feedback and the Area Chair for their efforts in coordinating the review process, which have been invaluable in improving our work.

Most reviewers recognized the novelty of our framework, its solid theoretical foundation, and the comprehensive experimental validation. During the rebuttal stage, we provided detailed point-by-point responses to all reviewer suggestions, including clarifying key formulations and addressing potential misunderstandings. We believe our responses have effectively addressed the concerns raised, and we are committed to incorporating these clarifications and improvements into the final version of the paper.

However, we have not yet received further feedback from the reviewers following our rebuttal. We are uncertain whether our responses have fully addressed their concerns, and we would be grateful for any additional guidance or clarification if needed.

Thank you once again for your support and thoughtful engagement throughout the review process.


Best regards,

The Authors

---

### Meta-Review · Area_Chair_EphP · 2026-01-04

**Summary:**

Reviewers raised concerns about inconsistencies in prior work descriptions, limited comparisons to recent baselines and mainstream LLMs, insufficient evaluation of explanation fidelity, hyperparameter sensitivity, potential reward hacking, and generalizability to novel relations or domains, which collectively question the paper's robustness, real-world applicability, and claims of SOTA performance, leading to a BORDERLINE rejected paper.

**Reviewer Concerns:**

The rebuttal addressed concerns on dictionary construction (via clarifications and model-agnostic experiments), baseline comparisons (by explaining task differences and adding LLM results), hyperparameter sensitivity (with new analyses showing stability), and reward hacking (through normalization mechanisms and ablation studies); however, outstanding issues include deeper validation of explanation fidelity beyond surface metrics, domain-specific testing for high-stakes applications

**Reviewer Scores:**

All reviewers would likely retain their original scores due to the clarity and incomplete comparisons of the current version.

---

### Decision · Program_Chairs · 2026-01-26

Reject